# LEARN WHAT YOU CAN'T LEARN: REGULARIZED ENSEMBLES FOR TRANSDUCTIVE OUT-OF-DISTRIBUTION DETECTION

## ABSTRACT

Machine learning models are often used in practice once they achieve good generalization results on in-distribution (ID) holdout data. To predict test sets in the wild, they should detect samples they cannot predict well. We show that current out-of-distribution (OOD) detection algorithms for neural networks produce unsatisfactory results in a variety of OOD detection scenarios, e.g. when OOD data consists of unseen classes or corrupted measurements. This paper studies how such "hard" OOD scenarios can benefit from tuning the detection method after observing a batch of the test data. This *transductive* setting is relevant when the advantage of even a slightly delayed OOD detection outweighs the financial cost for additional tuning. We propose a novel method that uses an artificial labeling scheme for the test data and early stopping regularization to obtain ensembles of models that produce contradictory predictions only on the OOD samples in a test batch. We show via comprehensive experiments that our approach is indeed able to significantly outperform both inductive and transductive baselines on difficult OOD detection scenarios, such as unseen classes on CIFAR-10/CIFAR-100, severe corruptions (CIFAR-C), and strong covariate shift ImageNet vs ObjectNet.

## 1 INTRODUCTION

Modern machine learning (ML) systems can achieve good test set performance and are gaining popularity in many real-world applications - from aiding medical diagnosis (Beede et al., 2020) to making recommendations for the justice system (Angwin et al., 2016). In reality however, some of the data points in a test set could come from a different distribution than the training (in-distribution) data. For example, sampling biases can lead to spurious correlations in the training set (Sagawa et al., 2020), a faulty sensor can produce novel data corruptions (Lu et al., 2019), or new unseen classes can emerge over time, like undiscovered bacteria (Ren et al., 2019). Many of these samples are so different compared to the training distribution that the model does not have enough information to predict their labels but still outputs predictions with high confidence. It is important to identify these out-of-distribution (OOD) samples in the test set and flag them, for example to at least temporarily abstain from prediction (Geifman & El-Yaniv, 2017) and involve a human in the loop.

To achieve this, Bayesian methods (Gal & Ghahramani, 2016; Malinin & Gales, 2018) or alternatives such as Deep Ensembles (Lakshminarayanan et al., 2017) try to identify samples on which a given model cannot predict reliably and include. Their aim is to obtain predictive models that simultaneously have low error on in-distribution (ID) data and perform well on OOD detection. Other approaches try to identify samples with low probability under the training distribution, independent of any prediction model, and use, for instance, density estimation (Nalisnick et al., 2019) or statistics of the intermediate layers of a neural network (Lee et al., 2018).

Most prior work have reported good OOD detection performance, reaching an almost perfect area under the ROC curve (AUROC) value of nearly 1. However these settings generally consider differentiating two vastly different data sets such as SVHN vs CIFAR10. We show that the picture is very different in a lot of other relevant settings. Specifically, for unseen classes within CIFAR10 or for data with strong distribution shifts (e.g. (resized) ImageNet vs ObjectNet (Barbu et al., 2019)), the AUROC of state-of-the-art methods often drops below 0.8.

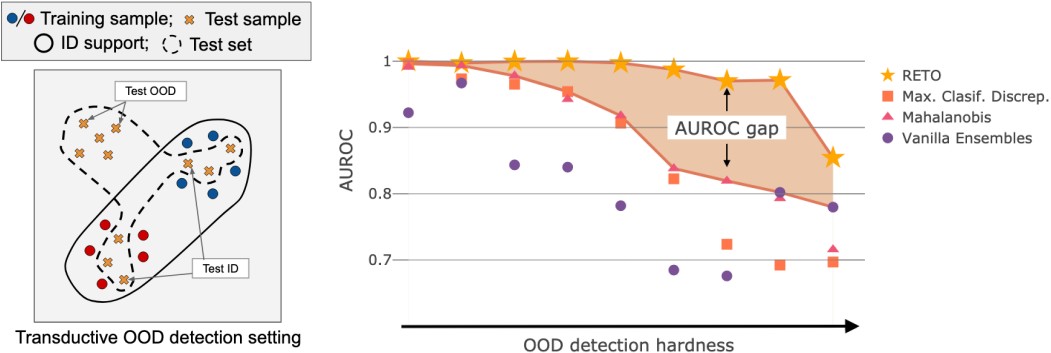

Figure 1: **Left:** Transductive OOD detection setting. Labeled training set and an unlabeled test set with samples both in- and outside the support of the training distribution. **Right:** Performance of RETO and some baselines on data sets ranked by their difficulty (Appendix C contains details on the hardness metric). The shaded area represents the gap in area under ROC curve (AUROC) between RETO and the next best baselines. The gap is wider for hard OOD detection settings.

Almost all of these methods assume a setting where at test time, no training is possible and the OOD detection method can only be trained beforehand. This *inductive setting* allows real-time decision-making and is hence more broadly used. However, in many cases we can indeed do batch predictions, for example when sensor readings come in every second and it is sufficient to make a prediction and decision every few minutes (e.g. automatic irrigation system). In this case we have a batch of unlabeled test data available that we want to predict (and be warned about) that we can use together with the labeled training set to detect the OOD points in the set. We call this the *transductive OOD setting* (related to but quite different from transductive classification (Vapnik, 1998)). Even in an online setting, transductive OOD could be very useful (see Section 2.1).

*(How) Can we achieve significantly better OOD detection performance in the transductive setting?*

Even though the transductive setting improves test accuracy in small data settings for tasks such as classification or zero-shot learning, it is unclear how to successfully leverage simultaneous availability of training and test set in the transductive OOD setting which is quite distinct from the former problems. A concurrent recent work Yu & Aizawa (2019) tackles this challenge by encouraging two classifiers to maximally disagree on the test set (i.e. to produce different predictions on test samples). However this leads to models that disagree to a similar degree on both ID and OOD data and hence one cannot distinguish between the two, as indicated by the low AUROC in Figure 1. We introduce a new method called Regularized Ensembles for Transductive OOD detection (RETO) for overparameterized models, which heavily uses regularization to make sure that the ensemble disagrees *only on the OOD* samples in the test set, but not on the ID samples. In summary, our main contributions in this paper are as follows:

- We experimentally identify many realistic OOD scenarios where SOTA methods achieve a subpar AUROC below $0.84$. We hence argue that the field of OOD detection is far from satisfactorily solved and more methods will be proposed that include these (or other) hard OOD cases as benchmarks.

- For the transductive OOD detection setting, we propose a new procedure, RETO, that manages to diversify the output of an ensemble only on the OOD portion of the test set and hence achieves significant improvements compared to SOTA methods (see Figure 1) with a relative gain of at least 32%.

## 2 REGULARIZED ENSEMBLES FOR TRANSDUCTIVE OOD DETECTION

Our main goal is to detect samples that are outside of the training distribution and focus on classification tasks. We are only interested in situations where we can obtain a model that generalizes well given the training data. If the models do not generalize well in-distribution (ID), then the primary task should be to find a better classifier instead. Given a classifier with good generalization, the next challenge becomes to ensure that samples on which the model cannot make confident predictions (e.g. samples that are too far from the training data) are correctly identified. This constitutes the

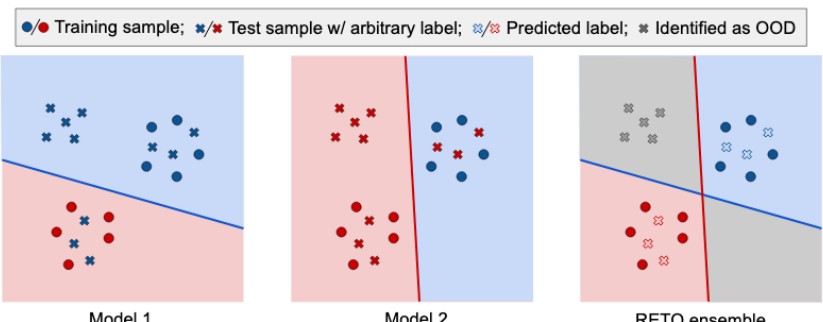

Figure 2: Cartoon illustration of an ensemble of two linear classifiers that fit both the training set and the test set. We assign either the blue label (Model 1) or the red label (Model 2) to the whole test set. Linear classifiers are *smooth enough* that they cannot fit both the correct labels of the training set and the arbitrary label on the test ID samples. The classifiers disagree only on the OOD points in the test set (gray crosses) and agree to predict the correct label on the test ID samples.

main focus of our work. Recall that in our use case, we have a batch of unlabeled test samples at our disposal. This test set includes a mixture of samples drawn from the training distribution and samples we call OOD per our definition in the previous section. The goal is to distinguish between the ID and the OOD samples in the test set.

In this section we propose our method that uses the more numerous training data as a counterweight that does not allow a sufficiently smooth model to fit an arbitrary label on ID test samples, but only on OOD test samples.

## 2.1 TRANSDUCTIVE OOD DETECTION

In an inductive OOD detection setting, one can only tune a method at training time, and then use it with unchanged parameters on any test set. In contrast, in a transductive setting, the training data is available during test time and it is possible to tune a method using both the training set and the unlabeled test set. We stress that no labels are available for the test data, so it is unknown which test samples are indeed anomalous. Moreover, we do not assume access to any *known* OOD samples, unlike some of the inductive methods, which sometimes use OOD data for training or calibration (Lee et al., 2018; Liang et al., 2018; Malinin & Gales, 2018; Cao et al., 2020).

When deployed in the context of classification, transductive and semi-supervised learning methods leverage the unlabeled data to obtain low-dimensional representations that are more effective for the prediction task. A key assumption for the setting to be useful is that the data is related in some way, e.g. the unlabeled data comes from the same distribution as the labeled data. On the other hand, transductive OOD detection differs from the usual transductive classification setting, in that the training distribution does not carry information about the OOD data. As a consequence, it is not obvious how to adapt existing semi-supervised methods to work in this different regime.

Some of the downsides that prevent transductive classification methods from being used more broadly are that for each test set that we want to predict, we would need to have access to the training data and computational resources. Furthermore, they do not allow predictions on the fly in the online setting. For transductive OOD however, we know that the inductive model predicts reliably in-distribution. Hence we can still predict test points on the fly, and *only flag OOD samples with a slight delay* after receiving a batch of test points. An example for which all these downsides are not limiting should be quite relatable to the reader. For example, Covid-19 test results have a crucial role for controlling the spread of the virus. Imagine a machine learning model were to be developed and deployed for reliable fast testing that works well under usual circumstances. If a test pipeline becomes defect, informing the patient of the potentially wrong test result is still crucial, in particular if it is to inform a negatively tested patient to repeat the test or to quarantine. In this case we would also be willing to allow access to labeled training data and computational resources for fine-tuning as precision is of utmost importance.

## 2.2 THE COMPLETE RETO PROCEDURE

We now provide details on our approach, RETO, outlined in Algorithm 1.

Recall that we have access to both a labeled training set, and the unlabeled test set. We begin by assigning an arbitrary label (selected from the set of labels of the training data) to all the test samples. We train a classifier on the union of the correctly-labeled training set, and the arbitrarily-labeled test set. To find the optimal classifier, we search among functions that are known to generalize well on the ID (training) distribution. If the classifiers are smooth enough, they will not be able to fit both the correct labels of the training set and the arbitrary label on the ID test samples, as illustrated in Figure 2 for linear classifiers. However, they will still fit the arbitrary label on the OOD test samples. Using regularization, we ensure that the models we obtain are not too complex. We search inside a function class of regularized functions, $\mathcal{F}_{\text{reg}}$, as discussed in more detail in Section 3. We ensemble several such classifiers, where each model fits a different label to the test set. We then use a disagreement statistic and flag as OOD all the points in the test set with high disagreement. To avoid training the ensemble from scratch for each new test batch, it is possible to instead start from pre-trained weights and perform a few iterations of fine-tuning, as detailed in Section 4. We stress that we do not calibrate or train our method on any known OOD data.

**Determining OOD samples with RETO.** We distinguish between ID and OOD samples using a two-sample statistical test, with the null hypothesis: $H_0 : x \in \text{supp}_P$, for a test sample $x$ and where $P$ denotes the training distribution[1]. Previous baselines have proposed their own choices of the test statistic, which is discussed in detail in Appendix A. For RETO, we use:

$$T_{\text{avg-TV}}(x) := \frac{1}{K(K-1)} \sum_{i \neq j} d_{\text{TV}}\left(f_i(x), f_j(x)\right),$$

the average pairwise total variation distance between the softmax outputs $f_i(x), f_j(x) \in \mathbb{R}^{|\mathcal{Y}|}$ of models $i, j \in \{1, ..., K\}$ in the ensemble, where $d_{TV}$ is the total variation distance. The null hypothesis is rejected for high values of $T_{\text{avg-TV}}$. Ap-

---

**Algorithm 1:** Pseudocode for RETO

**Input:** Train set $S$, Test set $T$, Ensemble size $K$,
   Test statistic threshold $t_0$, Regularized
   function class $\mathcal{F}_{\text{reg}}$, Disagreement metric

**Result:** $O$, i.e. the elements of $T$ which are OOD

**for** $c \leftarrow \{y_1, ..., y_K\}$ **do** // train $K$ models
  $T^c \leftarrow \{(x, c) : x \in T\}$
  $\hat{f}_c \leftarrow \textit{Train}\left(S \cup T^c; \mathcal{F}_{\text{reg}}\right)$
$O = \emptyset$
**for** $x \in T$ **do** // run two-sample test
  **if** $disagreement\left(\hat{f}_{y_1}(x), ..., \hat{f}_{y_K}(x)\right) > t_0$
  **then**
    $O \leftarrow O \cup \{x\}$

**return** $O$

---

pendix L contains more details about the choice of the test statistic. It follows from the way in which the hypothesis test is stated that true positives are OOD samples that are indeed flagged as OOD, while the false positives are ID samples that are incorrectly predicted as OOD.

## 3 WHY RETO CAN WORK WELL

In this section we provide insights as to why RETO can achieve a higher AUROC for hard OOD settings. We frame RETO as a way of learning diverse ensembles. We show that for RETO to perform well, the members of the ensemble must come from a restricted family of function classes. Finally, we argue that early-stopped neural networks are part of this restricted family.

**Ensemble-based OOD detection** The main argument for using an ensemble for OOD detection is that a diverse enough set of models will lead to disagreement only on OOD samples. In other words, the models will produce contradictory predictions on OOD inputs, while giving similar predictions for ID data. In order to get a diverse set of models, Deep Ensembles (Lakshminarayanan et al., 2017) use the stochasticity of the training process. However, the models obtained with this procedure are still very similar and tend to agree on OOD data (Figure 3), especially in hard OOD detection scenarios. Our method uses the additional information available in the unlabeled test set to generate ensembles that are more diverse on the OOD test samples. This approach allows the ensembles to work well even when the ID and OOD data are very similar.

We denote the test set as $T$ and consider a partition of it into a (unknown) test ID set $T_{\text{ID}}$ and a (unknown) test OOD set $T_{\text{OOD}}$. Recall that we do not know which test samples are outliers: if we

---

[1]With a slight abuse of notation, we denote by $\text{supp}_P \subset \mathcal{X}$ the support of the marginal distribution $P_X$.

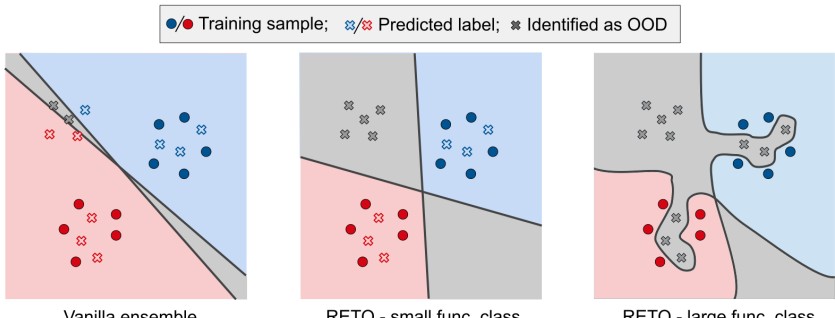

Figure 3: Comparison between different kinds of ensembles used for OOD detection. **Left:** A vanilla ensemble will not be diverse enough outside the training distribution support. **Middle:** In comparison, the RETO classifiers disagree only on the OOD points in the test set (gray crosses) and agree to predict the correct label on the test ID samples. **Right:** However, if the models in the ensemble are too complex (i.e. not regularized enough), then they can easily fit the arbitrary label on the ID test samples as well, thus identifying the whole test set as OOD.

were given $T = T_{\mathrm{OOD}}$, we could just enforce that different models learn different labels on $T$ by simply assigning one of the labels $c \in \mathcal{Y}$ to the whole test set to obtain $T^c := \{(x_i, c) : x_i \in T\}$, and then training each model in the ensemble on $S \cup T^c$, for different $c$. But, obviously, we are given the union of the test ID and test OOD set $T = T_{\mathrm{OOD}} \cup T_{\mathrm{ID}}$, without being able to distinguish between the two.

*How can we encourage an ensemble to disagree (only) on $T_{\mathrm{OOD}}$?*

Could we use the same strategy, and assign an arbitrary label to the entire test set? A neural network can easily learn random labels (Zhang et al., 2016). Hence, if we train a neural network to convergence on $S \cup T^c$, *the models will also disagree on the test ID data!* This is illustrated in Figure 3 - Right. How can we enforce models to have contradictory predictions on $T_{\mathrm{OOD}}$ but to *agree* on $T_{\mathrm{ID}}$?

### 3.1 KEY FOR TRANSDUCTIVE OOD DETECTION: REGULARIZATION

We can remedy this issue with strong regularization of the models in the ensemble. The *key intuition* is that it is difficult to fit an arbitrary label on ID data that is near the training samples. The signal in the correctly labeled ID points from the training set prevents an arbitrary label from being easily fit on the ID test samples. This is illustrated in Figure 2, for linear classifiers. Conversely, it is easy to learn the arbitrary label on samples that are far enough from the training data, which are exactly the OOD test samples! For instance, when training neural networks with SGD, the arbitrary label will be fit much faster on the OOD test samples than on the ID test samples.

*What is the right complexity for the models in the ensemble?*

In the language of statistical testing, the model complexity should be small enough to limit *false positives* (i.e. ID samples incorrectly flagged as OOD) and large enough to have enough *power* (i.e. identify correctly OOD samples). In other words, we want the classifiers to not fit the wrong labels on test ID samples, but only on test OOD samples. We encourage high power by making the models to fit different labels on the test set, and reduce false positives by *regularizing* the function class just enough. Specifically, we constrain our search to a hypothesis class that is just able to learn the labels on OOD, but not on ID test samples.

**Controlling false positives** We now describe a necessary condition on the complexity of the model class, in order to control the false positive rate of RETO. First of all, we require our models to generalize well in-distribution, since we they need to have enough common ground to agree on the test ID samples. If a model has poor generalization, then we should first find a better classifier before concerning ourselves with flagging OOD samples on which they cannot predict well.

However there may be model classes which generalize well but are still able to fit arbitrary labels on ID. Hence, we consider the set of *regularized functions with low population error*, i.e. $\mathcal{F}_\epsilon^\star := \{f \in \mathcal{F}_{\mathrm{reg}} : \mathbb{E}\left[\mathbb{1}_{f(x) \neq y}\right] < \epsilon\}$, where $\mathcal{F}_{\mathrm{reg}}$ is a restricted function class (e.g. functions parametrized by neural networks regularized via early-stopping). The required amount of regularization is captured in the following condition.

**Condition 3.1** ($\mathcal{F}_\epsilon^\star$ cannot fit noisy labels). *The probability of a point drawn from the ID distribution to be misclassified by a function in $\mathcal{F}_\epsilon^\star$ is at most $\delta$ for a small constant $\delta > \epsilon$.*

In words, the functions in $\mathcal{F}_\epsilon^\star$ only disagree on a small set with respect to the marginal distribution of $X$. As a consequence, with very high probability $1 - (1 - \delta)^s$ we cannot fit a set of $s$ random i.i.d. in-distribution points with the wrong label. This means the functions in $\mathcal{F}_\epsilon^\star$ are smooth enough to "ignore" wrong labels on in-distribution samples, as illustrated in Figure 3 - Middle. Note that unrestricted overparameterized models in the large function class $\mathcal{F}$ do not satisfy Condition 3.1 due to their capability to fit random labels (Figure 3 - Right).

**Notes on power** Note that Condition 3.1 enforces ensemble agreement on ID points to limit the amount of false positives. The power (ensemble disagreement on OOD) depends very much on the the the support of the OOD samples and its relation to $\mathcal{F}_\epsilon^\star$ (see Figure 4). If the boundary between the OOD and ID set requires higher complexity than $\mathcal{F}_\epsilon^\star$, our model class might have too little power as illustrated in Figure 4.

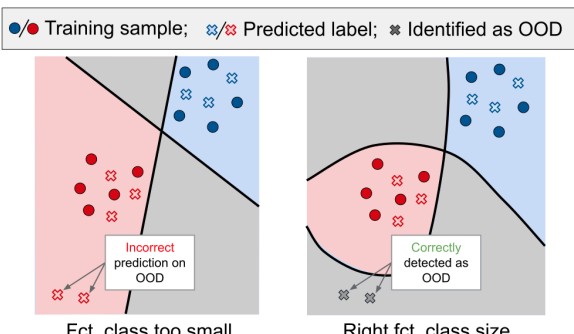

Figure 4: Importance of the right amount of regularization for power. **Left**: If the model class is too small, then we cannot fit the arbitrary label on test OOD points, and the classifiers will agree on them, leading to low statistical power. **Right**: In this situation, the OOD samples can only be detected using a larger function class.

The empirical OOD detection performance of our method indicates that early-stopping regularization finds just the right complexity - models that are complex enough for many hard OOD detection problems in image classification tasks that we consider. However, we leave as future work a thorough analysis of the trade-off between the function class size and the detection capabilities of RETO.

## 3.2 REGULARIZING NEURAL NETWORKS WITH EARLY STOPPING

An example of models that satisfy Condition 3.1 are deep neural networks trained with early stopping. The recent results of Yilmaz & Heckel (2019); Arora et al. (2019); Li et al. (2020) suggest that early stopping helps neural networks be more robust to label noise without sacrificing standard accuracy, thus satisfying Condition 3.1. Figure 5 shows the learning curves obtained when fitting a neural network on $S$, $T_{\mathrm{ID}}^c$ and $T_{\mathrm{OOD}}^c$ for a chosen label $c$: the training set and test OOD samples are fit first and after epoch 50, the predictor starts fitting the wrong label on the test ID set as well. We can also observe that early stopping at the point with the highest accuracy on a validation set (drawn from the same distribution as the training set), captures this phase transition well.

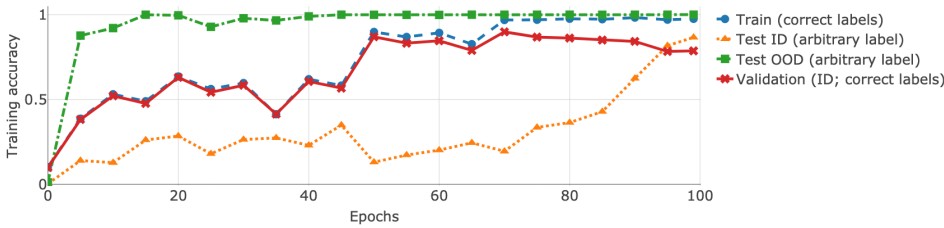

Figure 5: Accuracy measured on the correctly-labeled training set and on the arbitrarily-labeled test ID and OOD subsets. Validation accuracy is computed on a hold-out set with correctly-labeled ID samples. The training set and the test OOD samples are fit first, while the test ID set reaches high accuracy much later. At the early stopping iteration (after around 50 epochs) the models tend to predict the arbitrary label for test OOD samples and the correct label for test ID. The model is a Resnet20 trained on CIFAR10 as ID and SVHN as OOD.

## 4 EXPERIMENTS

In this section we evaluate the OOD detection performance of RETO for deep neural networks on several image data sets. We find that our approach outperforms all baselines on difficult OOD detection settings. In addition, we provide insights into the trade-off between offline detection and the good performance of our algorithm.

### 4.1 ID VS OOD SETTINGS

We report results on two broad types of OOD detection scenarios:

1. **Easy OOD data (most previous benchmarks):** ID and OOD samples come from very different distributions (e.g. CIFAR10 vs SVHN). These are the settings usually considered in the OOD detection literature on which most baselines perform well.

2. **Hard OOD data:** We explore two types of more difficult OOD detection tasks (i) The OOD data is sampled from "novel" classes: e.g. the first 5 digits of SVHN as training, the last 5 digits as OOD). (ii) The test data suffers from semi-strong covariate shift: e.g. the test set contains corrupted samples from the training distribution (e.g. CIFAR10 vs CIFAR10-C (Hendrycks & Dietterich, 2019) [2]) or samples that violate the spurious correlations present in the training set (e.g. ImageNet vs ObjectNet (Barbu et al., 2019)[3]).

Appendix C provides more insight on OOD detection hardness, while Appendix B presents examples of images for the various settings. Note that we are not too interested in the practical scenario of covariate shift (Shimodaira, 2000), where the distributions are so close that domain adaptation techniques could perform well.[4] In our hard OOD data sets, domain adaptation also leads to *unsatisfactory results*. For instance, Sun et al. (2020) obtain a classification error of 20.4% on CIFAR10-C at severity 5, compared to the 8.3% error achieved on the CIFAR10 test set. Alternatively, when domain adaptation fails, OOD detection can prompt a system to abstain on samples from the shifted distribution to prevent erroneous predictions.

Apart from using these canonical data sets, we also compare the performance of our method on more realistic data, namely a recently proposed OOD detection benchmark for medical imaging (Cao et al., 2020). The authors collected a suite of data sets that cover the aforementioned categories of difficulty, as detailed in Appendix K.

### 4.2 RETO VS. BASELINES

We compare our method against both inductive and transductive baselines. Importantly, some of the baselines require oracle knowledge of OOD data for training. For example, Outlier Exposure (Hendrycks et al., 2019) uses TinyImages for training as the set of outliers, irrespective of the OOD set used for evaluation. On the other hand, the *Mahalanobis* baseline (Lee et al., 2018) is tuned on samples from the same OOD distribution as the one seen at test time. We also present a transductive version of this approach, referred to as *Mahalanobis-T*, on which we elaborate in Appendix A.

For all the baselines we, use the default hyperparameters suggested by their authors and we do not adjust RETO for any of the OOD settings. We defer the details regarding training the models to Appendix A. For evaluation, we use two metrics that are common in the OOD detection literature: the area under the ROC curve (AUROC; larger values are better) and the false positive rate (FPR) at a true positive rate of 95% (FPR@95; smaller values are better).

### 4.3 MAIN RESULTS

For our method we train ensembles of five ResNet20 (He et al., 2016) networks (results for other architectures are presented in Appendix G). For each model in the ensemble we perform post-hoc

---

[2]Both CIFAR10-C and CIFAR100-C contain 15 types of corruptions, at 5 severity levels. We consider corrupted samples with severity 5.

[3]ObjectNet (Barbu et al., 2019) contains both novel classes that do not appear in ImageNet, and images from ImageNet classes, with strong distribution shift. We resize both ImageNet and ObjectNet to 32x32 images.

[4]These situations when domain adaptation performs well, are sometimes more challenging for OOD detection, since it means that ID and OOD data are more similar. In Appendix G.1 we show that RETO maintains its remarkable performance even on the more difficult CIFAR10-C data set with severity 2.

Table 1: Main results. We report the AUROC and the FPR@95. The **best RETO** and *best baseline* metrics are highlighted for each setting. For the corrupted CIFAR data sets, we report the average and the worst-case values, over all corruption types. For all instances, we used a training set of 40,000 labeled samples and a test set of size 20,000 with an equal number of ID and OOD samples.

| ID data | OOD data | kNN | DPN | Vanilla Ensembles Inductive | OE | Mahal. | Mahal-T | MCD | RETO (rand init) | RETO (pretrained) |
|---|---|---|---|---|---|---|---|---|---|---|
| | | | | | | | Transductive | | | |
| | | | | | AUROC ↑ / FPR@95 ↓ | | | | | |
| SVHN | CIFAR10 | 0.92 / 0.32 | *1.00 / 0.00* | 0.97 / 0.12 | *1.00 / 0.00* | 0.99 / 0.02 | 0.99 / 0.05 | 0.97 / 0.15 | **1.00 / 0.01** | 0.99 / 0.03 |
| CIFAR10 | SVHN | 0.81 / 0.74 | 0.95 / 0.15 | 0.92 / 0.22 | 0.97 / 0.11 | 0.99 / 0.04 | 0.99 / 0.04 | *1.00 / 0.02* | **1.00 / 0.00** | **1.00 / 0.00** |
| CIFAR100 | SVHN | 0.83 / 0.71 | 0.77 / 0.56 | 0.84 / 0.52 | 0.82 / 0.50 | *0.98 / 0.10* | *0.98 / 0.08* | 0.97 / 0.27 | **1.00 / 0.00** | **1.00 / 0.00** |
| SVHN[0:4] | SVHN[5:9] | 0.54 / 0.92 | 0.87 / 0.81 | *0.92 / 0.31* | 0.85 / 0.48 | *0.92 / 0.29* | 0.91 / 0.37 | 0.91 / 0.49 | **0.96 / 0.20** | 0.94 / 0.34 |
| CIFAR10[0:4] | CIFAR10[5:9] | 0.59 / 0.86 | *0.82 / 0.68* | 0.80 / 0.61 | *0.82 / 0.59* | 0.79 / 0.73 | 0.64 / 0.87 | 0.69 / 0.75 | **0.97 / 0.14** | 0.91 / 0.34 |
| CIFAR100[0:49] | CIFAR100[50:99] | 0.51 / 0.96 | 0.70 / 0.74 | *0.78 / 0.65* | 0.74 / 0.69 | 0.72 / 0.80 | 0.72 / 0.81 | 0.70 / 0.74 | **0.85 / 0.54** | 0.81 / 0.60 |
| CIFAR10 | CIFAR10-C (avg) | 0.67 / 0.72 | 0.89 / 0.40 | 0.84 / 0.51 | 0.86 / 0.46 | 0.94 / 0.20 | 0.88 / 0.37 | *0.95 / 0.16* | **1.00 / 0.00** | 1.00 / 0.01 |
| CIFAR10 | CIFAR10-C (worst) | 0.51 / 0.94 | 0.72 / 0.90 | 0.60 / 0.90 | 0.63 / 0.89 | *0.78 / 0.73* | 0.68 / 0.88 | 0.60 / 0.92 | **1.00 / 0.00** | 0.98 / 0.14 |
| CIFAR100 | CIFAR100-C (avg) | 0.67 / 0.73 | 0.74 / 0.64 | 0.78 / 0.63 | 0.76 / 0.63 | *0.92 / 0.28* | 0.84 / 0.45 | 0.91 / 0.35 | **1.00 / 0.01** | 0.99 / 0.03 |
| CIFAR100 | CIFAR100-C (worst) | 0.51 / 0.94 | 0.49 / 0.88 | 0.64 / 0.86 | 0.62 / 0.87 | *0.71 / 0.81* | 0.63 / 0.87 | 0.60 / 0.90 | **0.98 / 0.11** | 0.96 / 0.29 |
| Tiny ImageNet | Tiny ObjectNet | 0.51 / 0.96 | 0.70 / 0.68 | 0.82 / 0.51 | 0.79 / 0.63 | 0.75 / 0.74 | 0.72 / 0.75 | *0.99 / 0.02* | **0.99 / 0.02** | 0.98 / 0.12 |
| Average | | 0.67 / 0.77 | 0.83 / 0.52 | 0.85 / 0.45 | 0.85 / 0.46 | 0.89 / 0.36 | 0.85 / 0.42 | *0.90 / 0.33* | **0.98 / 0.10** | 0.96 / 0.16 |

early stopping: we train each model for 100 epochs and select the iteration with the lowest validation loss. For all settings, we used a labeled training set of 40,000 samples, a validation set of 10,000 ID samples and an unlabeled test set of 10,000 ID samples and 10,000 OOD samples. We present results for training the models from random initializations, and for fine-tuning pretrained models (pretraining is always performed on the training set for 100 epochs). When using pretrained weights, as few as three epochs of fine-tuning are enough on average to achieve the performance that we report, which is a significant cut in computation cost. In addition, Appendix D shows the dependence on the ensemble size for RETO and vanilla ensemble.

Table 1 summarizes the main empirical results. For the corruption data sets, the table shows the average of the AUROC and FPR@95 taken over all corruptions, and the value for the worst-case setting. Appendix G contains a more detailed breakdown of these numbers. In addition to being successful at identifying OOD samples, our method also maintains a good classification performance on the training distribution. The validation accuracy of the early stopped ensembles, averaged over all settings, is only 1.4% smaller than that of vanilla ensembles.

The evaluation for the scenarios presented in Table 1 is performed on the same test set that was used for training, as usual in transductive learning. In addition to that, the OOD detection performance of RETO extrapolates well to unseen samples from the same distribution.[5] In order to show this, we run experiments in which we compute the AUROC on a hold-out test set drawn from the same ID and OOD distributions as the one used during training. The AUROC on the hold-out test set is within 0.01 from the one calculated on the test set observed during training.

For the medical OOD detection benchmark we present the average AUROC achieved by some representative baselines in Figure 6a. We refer the reader to Cao et al. (2020) for precise details on the methods. Appendix K contains more results for the medical settings, as well as additional baselines.

**Limitations and trade-offs.** Having access to the test set in the transductive setting provides enough information to discriminate well between the ID and the OOD test samples, succeeding when inductive approaches are less effective. In order to bridge the gap between (offline) transductive OOD detection and online anomaly detection we investigate the impact of the size of the test set on the OOD detection performance. In addition, we also vary the ratio of OOD samples in the test set, i.e. $\frac{|T_{\mathrm{OOD}}|}{|T_{\mathrm{ID}}|+|T_{\mathrm{OOD}}|}$. Our findings suggest that there is a broad spectrum of values for which RETO maintains a good performance. In the cases when either the size of the test set or the test OOD ratio is small, the OOD detection performance deteriorates to the point where it is comparable to vanilla ensembles, as shown in Figure 6b where we report the gap in AUROC between RETO and a vanilla ensemble. When there are only a few very diverse OOD test samples, their contribution to the gradient is small. Moreover, if the number of ID test samples is large, fitting a single arbitrary

---

[5]This setting is similar for instance to the one in the Mahalanobis baseline, which assumes oracle knowledge of the OOD distribution at training time.

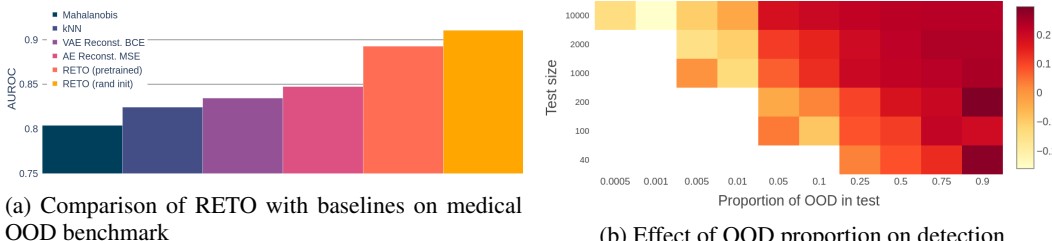

(a) Comparison of RETO with baselines on medical OOD benchmark

(b) Effect of OOD proportion on detection

Figure 6: **Left:** AUROC averaged over all scenarios in the medical OOD detection benchmark. The values for the baselines are computed using the code from the paper Cao et al. (2020). **Right:** The gap in AUROC between RETO and a vanilla ensemble, as the number and proportion of ID (CIFAR10) and OOD (CIFAR10-C/snow) samples in the test set is varied. The AUROCs are obtained with an ensemble of ResNet20 models.

label on them is significantly easier. This, combined with the signal in the OOD test data being weaker, means that the arbitrary label can take longer to fit on the few OOD samples, than on the (numerous) test ID samples, leading to many false positives (i.e. ID samples incorrectly flagged as OOD). The loss in efficacy can be mitigated by either splitting the test set in smaller batches, or by using a different labeling scheme for the test set, the details of which we leave as future work.

## 5 RELATED WORK

**Transductive learning.** Transductive learning (Vapnik, 1998) has been successfully used for practical applications in problems like zero-shot learning (Fu et al., 2015; Kodirov et al., 2015; Wan et al., 2019). Scott & Blanchard (2008) proposed to solve the transductive anomaly detection problem simply by discriminating between the ID and OOD distribution with a constraint on the false positive rate. However, it is difficult to assess the predictive uncertainty of a classifier trained on the source set, using this binary classifier. The setting we consider in which one has access to both a labeled and an unlabeled data set, is reminiscent of the problem of **semi-supervised learning**. Approaches based on self-trained predictors have recently been proposed by Kumar et al. (2020); Chen et al. (2020); Sun et al. (2020) in the context of domain adaptation.

**Ensemble diversity.** Some limitations of ensemble-based OOD detection approaches have been highlighted in recent years. First, neural networks tend to make overconfident predictions even on test samples far from the training distribution (Hein et al., 2019). Moreover, the stochasticity of conventional NN training approaches (e.g. random initialization, SGD) is not sufficient to obtain ensembles that are diverse enough to give good uncertainty estimates on OOD samples (Maennel & Tifrea, 2020). Some recent works address this problem by adding explicit regularizers that incentivize model diversity, either in a transductive setting, like MCD (Yu & Aizawa, 2019), or inductively (Bahng et al., 2020), but do not manage to detect OOD samples well in hard scenarios.

**Bayesian prediction.** One of the important appeals of the Bayesian framework is that it directly provides uncertainty estimates together with the predictions, in the form of a posterior distribution. Approaches like MC-Dropout (Gal & Ghahramani, 2016) or Deep Prior Networks (Malinin & Gales, 2018) have been proposed in the context of OOD detection, but the uncertainty estimates they provide are often inaccurate on OOD samples (Ovadia et al., 2019). Even though Bayesian Neural Networks (Neal, 1996) have seen some progress in recent years (Graves, 2011; Blundell et al., 2015), sampling efficiently from the posterior over parameters remains an important open problem.

## 6 CONCLUSIONS

Reliable OOD detection is essential in order for classification systems to be deployed in safety-critical environments. We present a method that achieves state-of-the-art performance in a transductive OOD detection setting, and which, like other approaches, can ultimately be used to abstain on OOD samples. As future work, we propose a more thorough investigation of the influence of the labeling scheme of the test set on the sample complexity of the method, as well as an analysis of the trade-off governed by the complexity of the model class of the classifiers.

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

# A    EXPERIMENT DETAILS

## A.1    BASELINES

We instantiate all baselines with the hyperparameters suggested by the authors for the respective settings (e.g. different hyperparameters for CIFAR10 or ImageNet). For all methods, we use pre-trained models provided by the authors when available and we pre-train our own models when that is not the case. When doing our own pre-training, we always use the parameters described in the original paper. The code published for the *Mahalanobis* method performs a hyperparameter search automatically for each of the settings on which we ran it.

- **k-Nearest Neighbors**: We take $k = 8$. For each test sample, we take the average distance to the nearest neighbors in the input (pixel) space, and we use this as the test statistic.

- **Vanilla Ensembles** (Lakshminarayanan et al., 2017): We train an ensemble on the training set according to the true labels. For a test sample, average the models' probabilities, and use the entropy of the resulting distribution as the test statistic. We use ensembles of 5 models, with the same architecture and hyperparameters as the ones used for RETO.

- **Outlier Exposure** (Hendrycks et al., 2019): It makes the model's softmax predictions close to the uniform distribution on the known outliers, while maintaining a good classification performance on the training distribution. We use the WideResNet (Zagoruyko & Komodakis, 2016) for the RGB data sets. For fine-tuning, we use their recommended settings of 10 epochs at learning rate 0.001. For training from scratch, we train for 100 epochs with an initial learning rate of 0.1. When the training dataset is either CIFAR and ImageNet, we use the default WRN parameters of the author's code, namely 40 layers, 2 widen-factor, droprate 0.3. For when the training dataset is SVHN, we use the author's recommended parameters of 16 layers, 4 widen-factor and droprate 0.4. All settings use the cosine annealing learning rate scheduler provided with the author's code, without any modifications.

- **Deep Prior Networks (DPN)** (Malinin & Gales, 2018): Bayesian Method: It trains a neural network (Prior Network) to parametrize a Dirichlet distribution over the class probabilities. We train for 100 epochs an WRN-28-10 using SGD with momentum 0.9, with an initial learning rate of 0.01, which is decayed by 0.2 at epochs 50, 70, and 90. For MNIST, we use EMINST/Letters as OOD for tuning. For all other settings, we use TinyImages as OOD for tuning.

- **Mahalanobis** (Lee et al., 2018): It pretrains models on the training data. For a data point, it uses the intermediate representations of each layer as "extracted features". It then performs binary classification using logistic regression using these extracted features. In the original setting, the classification is done on "training" ID vs "training" OOD samples (which are from the same distribution as the test OOD samples). Furthermore, hyperparameter tuning for the optimal amount of noise is performed on "validation" ID and OOD data. We use the WRN-28-10 architeture, pretrained for 200 epochs. The initial learning rate is 0.1, which is decayed at epochs 60, 120, and 160 by 0.2. We use SGD with momentum 0.9, and the standard weight decay of $5 \cdot 10^{-4}$.

- **Mahalanobis-Transductive**: The methodology proposed by Lee et al. (2018) is very different from the other settings, where we do not have access to samples which are known to be OOD and from the same distribution as test OOD. Therefore, we propose a transductive alternative: early-stopped logistic regression is used to distinguish between the training set and the test set (instead of ID vs OOD samples). The early stopping iteration is chosen to minimize the classification errors on a validation set that contains only ID data (recall that we do not assume to know which are the OOD samples).

- **Maximum Classifier Discrepancy (MCD)** (Yu & Aizawa, 2019): It is a transductive method that trains two classifiers at the same time, and makes them disagree on the test data, while maintaining good classification performance. We use the WRN-28-10 architecture as suggested in the paper. We did not change the default parameters which came with the author's code, so weight decay is $10^{-4}$, and the optimizer is SGD with momentum 0.9. When available (for CIFAR10 and CIFAR100), we use the pretrained models provided

by the authors. For the other training datasets, we use their methodology to generate pre-trained models: We train a WRN-28-10 for 200 epochs. The learning rate started at 0.1 and dropped by a factor of 10 at $50\%$ and $75\%$ of the training progress, respectively.

## A.2    Training configuration for regularized ensembles

When training RETO with a certain neural network architecture, we use hyperparameters that give the best test accuracy when training a model on the ID training set. We do not perform further hyperparameter tuning for the different OOD data sets on which we evaluate our approach.

For MNIST, we train a 3-layer MLP with ReLU activations. Each intermediate layer has 100 neurons. The model is optimized using Adam, using a learning rate of $0.001$, for 10 epochs.

For CIFAR and ImageNet, we train a WideResNet WRN-28-10. The model is trained using SGD with momentum 0.9, and the learning rate starts at $0.1$, and is multiplied by $0.2$ at epochs 50, 70 and 90. The weights have a l2 regularization coefficient of $5e - 4$. We use a batch size of 128 for all scenarios. The hyperparameters have been selected to achieve high accuracy on the CIFAR100 classification problem, thus obtaining an ensemble validation accuracy of $80.8\%$, where each individual model has between $77\%$ and $78\%$ accuracy. After that we used the same hyperparameters for all settings. For the fine-tuning scenarios, we trained for 10 epochs with a constant learning rate of 0.001 for all scenarios.

For the medical datasets, we train a Densenet-121 as the authors do in the original paper. For training from scratch, we do not use random weight initializations, but instead we start with the ImageNet weights provided with tensorflow. The training configuration is exactly the same as for WRN-28-10, except for: we use a batch size of 32 because of GPU memory restrictions, and for fine tuning we use a constant learning rate of $10^{-5}$.

# B    ID and OOD data sets

## B.1    Data sets

For evaluation, we use the following image data sets:

- MNIST (Lecun et al., 1998) and Fashion MNIST (Xiao et al., 2017).

- SVHN (Netzer et al., 2011).

- CIFAR10, CIFAR100 (Krizhevsky, 2009), and their corrupted variants (Hendrycks & Dietterich, 2019).

- ImageNet (Deng et al., 2009) and ObjectNet (Barbu et al., 2019), both resized to 32x32.

## B.2 Samples for the settings with novel classes

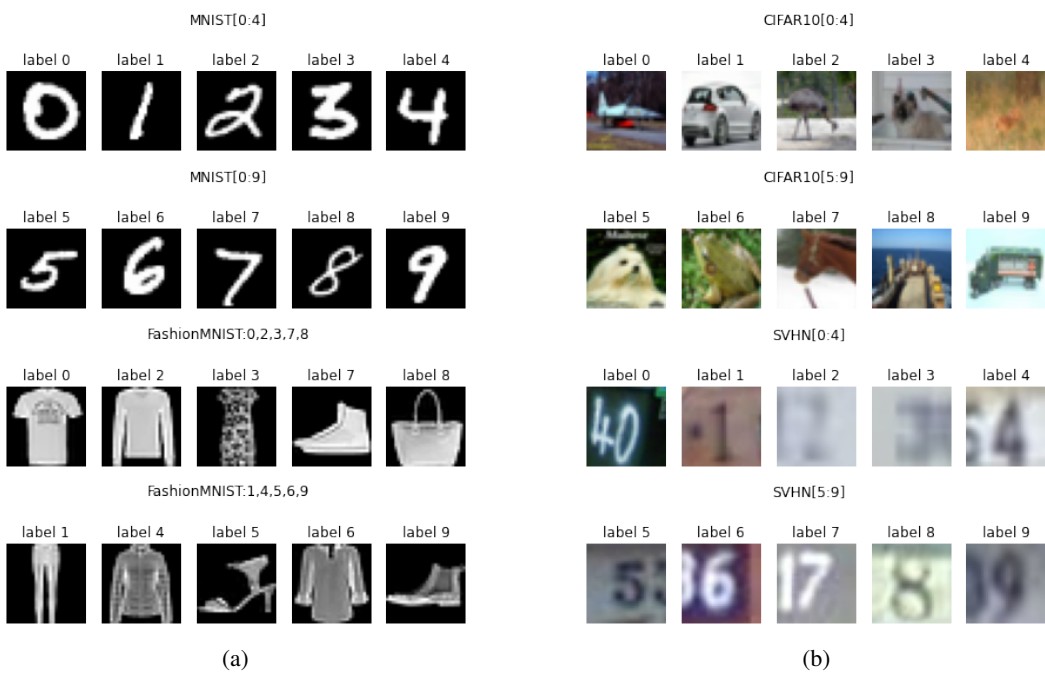

(a)                                                                (b)

Figure 7: (a) Data samples for the MNIST/FashionMNIST splits. (b) Data samples for the CIFAR10/SVHN splits.

## B.3 Samples from ObjectNet

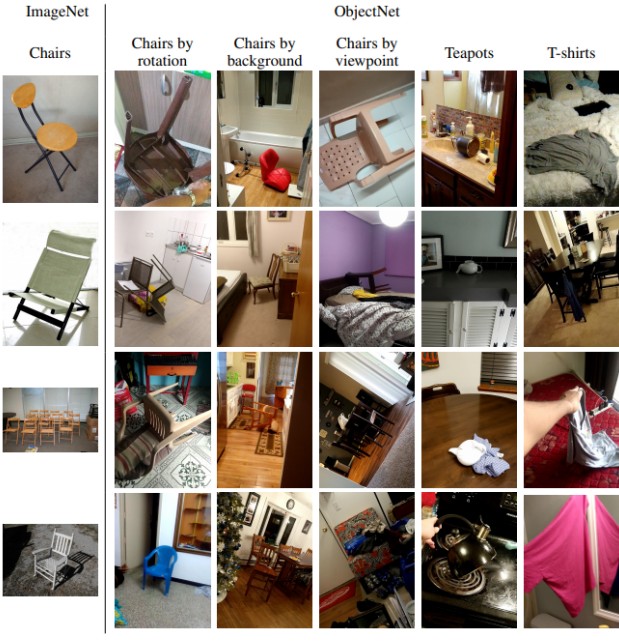

Figure 8: Samples from ImageNet and ObjectNet taken from the original paper by Barbu et al. (2019).

### B.4 SAMPLES FROM CIFAR10-C

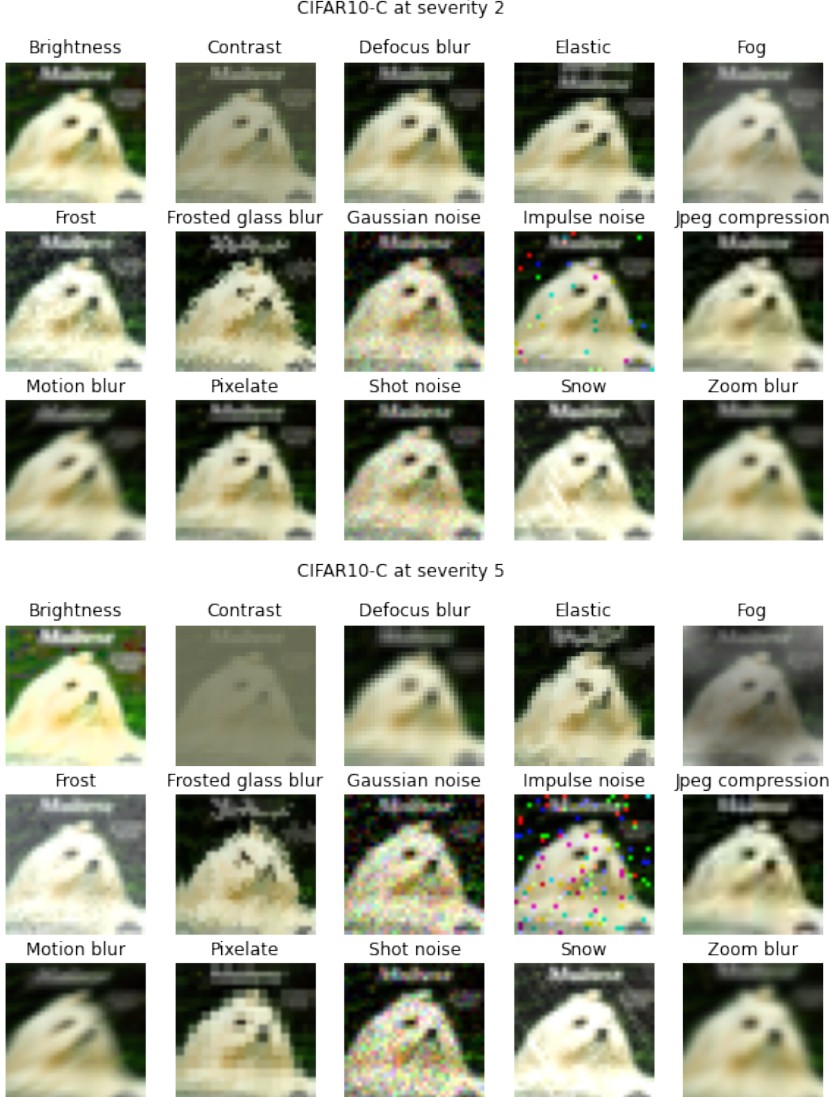

Figure 9: Data samples for the corrupted CIFAR10-C data set.

## C OOD DETECTION HARDNESS

Out of distribution detection benchmarks can be assessed based on their difficulty. In what follows, we propose a simple way to evaluate the hardness of an OOD detection setting and provide empirical evidence that shows that the scenarios we looked at are indeed more complicated than some of the common OOD detection benchmarks.

Consider the task of distinguishing between samples that come from two distributions $P, Q$ with disjoint supports $\text{supp}_P, \text{supp}_Q$. Let us assign labels according to the distribution the points are coming from: $\mathcal{D} = \{(x_i, y_i) : y_i = -1 \text{ if } x_i \in \text{supp}_P, y_i = 1 \text{ if } x_i \in \text{supp}_Q\}$. We search solve the classification problem by searching for a minimizer of the empirical risk inside a function class $\mathcal{F}$.

The intuition for our measure of *hardness* is as follows: if it is difficult for a binary classifier to separate samples from $P$ and $Q$, then it will also be difficult to detect test samples from $Q$ as OOD, when only a training set drawn from $P$ is available.

To quantify the difficulty of the binary classification problem, we use the area under the training curve, i.e. the curve of the training loss as a function of iterations of the optimization algorithm. The larger the area, the more iterations it takes to converge, which in turn indicates that the classification problem is difficult.

Formally, we define the *hardness* of the OOD detection task with respect to a function class $\mathcal{F}$ as:

$$\mathcal{H}_{OODD}(\mathcal{D}; \mathcal{F}) := \int_0^1 L(f_t)dt,$$

where $f_t$ is the model after a fraction $t$ of the training epochs are finished.

For our task, we start with a VGG model and train it for 30 epochs. In order to approximate the integral, we take the training loss for the whole data set every 5 epochs, and we average these losses.

Table 2: OOD detection hardness.

|  | **ID data set** | **OOD data set** | $\mathcal{H}_{OODD}$ |
|---|---|---|---|
| 1) Easy OOD | MNIST | FashionMNIST | 0.01 |
|  | FashionMNIST | MNIST | 0.01 |
|  | SVHN | CIFAR10 | 0.05 |
|  | CIFAR10 | SVHN | 0.05 |
|  | CIFAR100 | SVHN | 0.06 |
| 2) Novel classes | MNIST[0:4] | MNIST[5:9] | 0.15 |
|  | FashionMNIST:0,2,3,7,8 | FashionMNIST:1,4,5,6,9 | 0.27 |
|  | SVHN[0:4] | SVHN[5:9] | 0.24 |
|  | CIFAR10[0:4] | CIFAR10[5:9] | 0.45 |
|  | CIFAR100[0:49] | CIFAR100[50:99] | 0.59 |
| 3) Hard covariate shift | CIFAR10 | CIFAR10-C sev5 | 0.08 |
|  | CIFAR100 | CIFAR100-C sev5 | 0.10 |
|  | ImageNet | ObjectNet | 0.18 |
| 3') Easy covariate shift | CIFAR10 | CIFAR10-C sev2 | 0.18 |
|  | CIFAR100 | CIFAR100-C sev2 | 0.19 |

Notice that the settings with novel classes and the one with hard covariate shift are generally more difficult than the common benchmarks used in the OOD detection literature.

Apart from the three categories of settings that we introduced in Section 4, we also present numbers for CIFAR10-C and CIFAR100-C with lower-severity corruptions, i.e. severity 2. These scenarios are usually easier to solve with domain adaptation techniques. Nevertheless, in Appendix G we show that RETO performs well on OOD detection on these settings as well. Even though performing OOD detection is redundant here, the good results of our method go to show that it can still work well, even in difficult situations.

# D  RESOURCE REQUIREMENTS FOR RETO

**Computational cost**  Our method can work with training each model in the ensemble from scratch (i.e. random initialization), but it also preforms well when fine-tuning a network from pretrained weights. This reduces significantly the inference time for each batch of test data. For the settings we considered, on average, as few as three epochs of fine tuning are enough to achieve the best performance: the training is stopped early, on average, after three epochs, according to the condition on the validation loss.

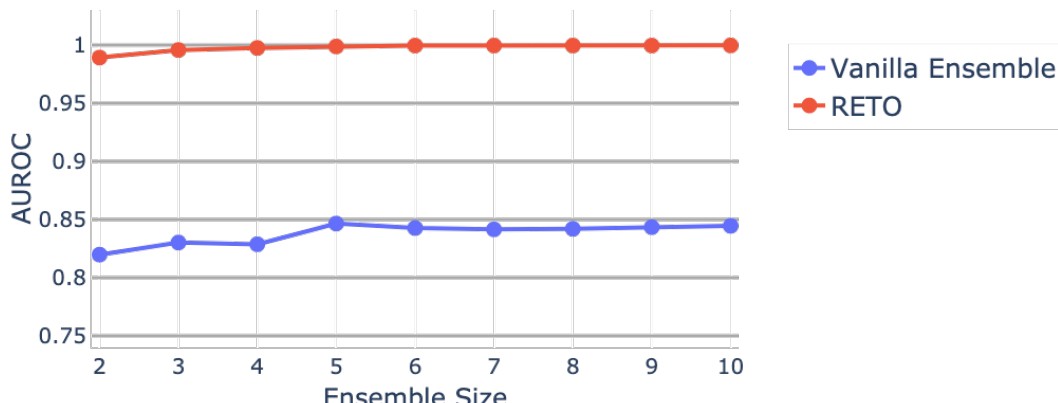

Figure 10: Effect of ensemble size on CIFAR100 vs. SVHN. Both methods are trained from scratch for 100 epochs.

**Dependence on ensemble size**  Figure 10 shows that the good performance of RETO does not rely on a large number of models in the ensemble. Unlike vanilla ensembles, our method achieves a high AUROC with as few as 2 models. This is because, in vanilla ensembles, the networks are diverse 'by chance', due to the stochasticity of the training procedure. On the other hand, in RETO, our training method actively encourages the ensembles to be diverse, and hence two models will already disagree on OOD data almost as much as five.

# E  GENERALIZATION TO HOLD-OUT TEST SET

In this section we present experiments which show that after training/fine-tuning on a test set with ID and OOD samples, one can also use our method to detect OOD samples from the same distribution, that have not been seen during training. Concretely, we use a test set of 5000 ID and 5000 OOD samples to train RETO ensembles (we reiterate that we do not have access to which samples are indeed OOD in the test set). For evaluation, we compute the metrics on a separate data set, with 5000 ID and 5000 OOD samples, where the OOD samples come from the same distribution as the samples seen during training. As revealed in Table 3, the performance does not change substantially, when evaluating on the hold-out test set. For the corruption data sets, we report for each metric the average taken over all corruptions (A), and the value for the worst-case setting (W).

Table 3: Generalization on held out test set.

| ID data | OOD data | RETO (normal) | RETO (holdout) |
|---|---|---|---|
| | | AUROC ↑ / FPR@95 ↓ | |
| SVHN | CIFAR10 | 0.99 / 0.03 | 0.99 / 0.02 |
| CIFAR10 | SVHN | 1.00 / 0.00 | 1.00 / 0.00 |
| CIFAR100 | SVHN | 1.00 / 0.00 | 1.00 / 0.00 |
| SVHN[0:4] | SVHN[5:9] | 0.94 / 0.34 | 0.95 / 0.28 |
| CIFAR10[0:4] | CIFAR10[5:9] | 0.91 / 0.34 | 0.90 / 0.37 |
| CIFAR100[0:49] | CIFAR100[50:99] | 0.81 / 0.60 | 0.79 / 0.63 |
| CIFAR10 | CIFAR10-C sev 2 (A) | 0.96 / 0.14 | 0.95 / 0.16 |
| CIFAR10 | CIFAR10-C sev 2 (W) | 0.68 / 0.81 | 0.65 / 0.84 |
| CIFAR10 | CIFAR10-C sev 5 (A) | 1.00 / 0.01 | 1.00 / 0.02 |
| CIFAR10 | CIFAR10-C sev 5 (W) | 0.98 / 0.14 | 0.97 / 0.21 |
| CIFAR100 | CIFAR100-C sev 2 (A) | 0.94 / 0.24 | 0.95 / 0.23 |
| CIFAR100 | CIFAR100-C sev 2 (W) | 0.71 / 0.81 | 0.71 / 0.84 |
| CIFAR100 | CIFAR100-C sev 5 (A) | 0.99 / 0.03 | 0.99 / 0.03 |
| CIFAR100 | CIFAR100-C sev 5 (W) | 0.96 / 0.29 | 0.95 / 0.30 |
| | Average | 0.97 / 0.11 | 0.97 / 0.11 |

# F  REGULARIZED TRAINING/TEST DISCRIMINATOR FOR TRANSDUCTIVE OOD DETECTION

The paper Scott & Blanchard (2008) suggests that training a binary classifier with bounded false positive rate to distinguish between the training set $S$ and the test set $T$ can successfully separate the OOD samples from the ID samples in the test set. However, this approach does not fall in the category of predictive uncertainty-based OOD detection methods, since it does not provide a good classifier of the labeled training set as well.

We present in what follows a set of experiments run to check if a similar technique works for the data sets we considered. Early stopping with respect to a validation set that contains only ID samples is enough to obtain good OOD detection performance.

For the corruption data sets, the table shows the average of the AUROC taken over all corruptions (A), and the value for the worst-case setting (W).

Table 4: AUROC for an early-stopped binary classifier trained to separate the training set from the test set.

| ID data | OOD data | Binary classifier AUROC ↑ |
|---|---|---|
| SVHN | CIFAR10 | 0.99 |
| CIFAR10 | SVHN | 1.00 |
| CIFAR100 | SVHN | 1.00 |
| SVHN[0:4] | SVHN[5:9] | 0.82 |
| CIFAR10[0:4] | CIFAR10[5:9] | 0.83 |
| CIFAR100[0:49] | CIFAR100[50:99] | 0.61 |
| CIFAR10 | CIFAR10-C sev 2 (A) | 0.94 |
| CIFAR10 | CIFAR10-C sev 2 (W) | 0.51 |
| CIFAR10 | CIFAR10-C sev 5 (A) | 1.00 |
| CIFAR10 | CIFAR10-C sev 5 (W) | 1.00 |
| CIFAR100 | CIFAR100-C sev 2 (A) | 0.83 |
| CIFAR100 | CIFAR100-C sev 2 (W) | 0.50 |
| CIFAR100 | CIFAR100-C sev 5 (A) | 1.00 |
| CIFAR100 | CIFAR100-C sev 5 (W) | 1.00 |
| Tiny ImageNet | Tiny ObjectNet | 1.00 |
| Average | | 0.94 |

# G MORE EXPERIMENTS

## G.1 EXTENDED RESULTS WITH RESNET

Table 5: Extended results on all setting (including a breakdown for all corruption types). For RETO and vanilla ensembles, we train 5 ResNet20 models for each setting.

| ID data | OOD data | kNN | DPN | Vanilla Ensembles | OE | Mahal. | Mahal-T | MCD | RETO (rand init) | RETO (pretrained) |
|---|---|---|---|---|---|---|---|---|---|---|
| | | | | | AUROC ↑ / FPR@95 ↓ | | | | | |
| SVHN | CIFAR10 | 0.92 / 0.32 | 1.00 / 0.00 | 0.97 / 0.12 | 1.00 / 0.00 | 0.99 / 0.02 | 0.99 / 0.05 | 0.97 / 0.15 | 1.00 / 0.01 | 0.99 / 0.03 |
| CIFAR10 | SVHN | 0.81 / 0.74 | 0.95 / 0.15 | 0.92 / 0.22 | 0.97 / 0.11 | 0.99 / 0.04 | 0.99 / 0.04 | 1.00 / 0.02 | 1.00 / 0.00 | 1.00 / 0.00 |
| CIFAR100 | SVHN | 0.83 / 0.71 | 0.77 / 0.56 | 0.84 / 0.52 | 0.82 / 0.50 | 0.98 / 0.10 | 0.98 / 0.08 | 0.97 / 0.27 | 1.00 / 0.00 | 1.00 / 0.00 |
| SVHN[0:4] | SVHN[5:9] | 0.54 / 0.92 | 0.87 / 0.81 | 0.92 / 0.31 | 0.85 / 0.48 | 0.92 / 0.29 | 0.91 / 0.37 | 0.91 / 0.49 | 0.96 / 0.20 | 0.94 / 0.34 |
| CIFAR10[0:4] | CIFAR10[5:9] | 0.59 / 0.86 | 0.82 / 0.68 | 0.80 / 0.61 | 0.82 / 0.59 | 0.79 / 0.73 | 0.64 / 0.87 | 0.69 / 0.75 | 0.97 / 0.14 | 0.91 / 0.34 |
| CIFAR100[0:49] | CIFAR100[50:99] | 0.51 / 0.96 | 0.70 / 0.74 | 0.78 / 0.65 | 0.74 / 0.69 | 0.72 / 0.80 | 0.72 / 0.81 | 0.70 / 0.74 | 0.85 / 0.54 | 0.81 / 0.60 |
| CIFAR10 | CIFAR10-C brightness 2 | 0.54 / 0.93 | 0.55 / 0.93 | 0.51 / 0.95 | 0.52 / 0.94 | 0.58 / 0.92 | 0.55 / 0.93 | 0.52 / 0.94 | 0.86 / 0.59 | 0.68 / 0.81 |
| CIFAR10 | CIFAR10-C contrast 2 | 0.94 / 0.20 | 0.69 / 0.86 | 0.59 / 0.91 | 0.62 / 0.89 | 0.82 / 0.56 | 0.69 / 0.79 | 0.82 / 0.91 | 1.00 / 0.00 | 0.97 / 0.17 |
| CIFAR10 | CIFAR10-C defocus blur 2 | 0.56 / 0.91 | 0.48 / 0.95 | 0.54 / 0.93 | 0.56 / 0.93 | 0.67 / 0.81 | 0.58 / 0.90 | 0.60 / 0.90 | 0.99 / 0.07 | 0.94 / 0.32 |
| CIFAR10 | CIFAR10-C elastic 2 | 0.56 / 0.92 | 0.47 / 0.97 | 0.63 / 0.87 | 0.65 / 0.87 | 0.71 / 0.76 | 0.66 / 0.85 | 0.68 / 0.91 | 0.99 / 0.08 | 0.93 / 0.37 |
| CIFAR10 | CIFAR10-C fog 2 | 0.81 / 0.52 | 0.61 / 0.89 | 0.53 / 0.93 | 0.55 / 0.93 | 0.81 / 0.62 | 0.71 / 0.73 | 0.58 / 0.92 | 0.99 / 0.06 | 0.90 / 0.39 |
| CIFAR10 | CIFAR10-C frost 2 | 0.50 / 0.92 | 0.84 / 0.60 | 0.69 / 0.84 | 0.69 / 0.85 | 0.90 / 0.42 | 0.72 / 0.73 | 0.59 / 0.95 | 1.00 / 0.00 | 1.00 / 0.01 |
| CIFAR10 | CIFAR10-C frosted glass blur 2 | 0.51 / 0.95 | 0.98 / 0.09 | 0.89 / 0.42 | 0.86 / 0.50 | 0.99 / 0.02 | 0.99 / 0.04 | 0.99 / 0.04 | 1.00 / 0.00 | 1.00 / 0.00 |
| CIFAR10 | CIFAR10-C gaussian noise 2 | 0.57 / 0.89 | 0.97 / 0.13 | 0.83 / 0.62 | 0.85 / 0.58 | 1.00 / 0.01 | 0.99 / 0.02 | 1.00 / 0.00 | 1.00 / 0.00 | 1.00 / 0.00 |
| CIFAR10 | CIFAR10-C impulse noise 2 | 0.61 / 0.85 | 0.95 / 0.23 | 0.80 / 0.68 | 0.81 / 0.61 | 1.00 / 0.02 | 0.99 / 0.04 | 1.00 / 0.00 | 1.00 / 0.00 | 1.00 / 0.00 |
| CIFAR10 | CIFAR10-C jpeg compression 2 | 0.51 / 0.95 | 0.65 / 0.92 | 0.71 / 0.81 | 0.73 / 0.77 | 0.75 / 0.74 | 0.68 / 0.84 | 1.00 / 0.00 | 1.00 / 0.00 | 1.00 / 0.00 |
| CIFAR10 | CIFAR10-C motion blur 2 | 0.60 / 0.88 | 0.65 / 0.88 | 0.71 / 0.81 | 0.74 / 0.77 | 0.80 / 0.69 | 0.79 / 0.61 | 0.99 / 0.01 | 1.00 / 0.00 | 0.99 / 0.02 |
| CIFAR10 | CIFAR10-C pixelate 2 | 0.52 / 0.94 | 0.82 / 0.67 | 0.65 / 0.88 | 0.66 / 0.85 | 0.79 / 0.71 | 0.66 / 0.86 | 0.93 / 0.73 | 1.00 / 0.00 | 0.99 / 0.04 |
| CIFAR10 | CIFAR10-C shot noise 2 | 0.53 / 0.92 | 0.87 / 0.61 | 0.74 / 0.79 | 0.75 / 0.82 | 0.98 / 0.06 | 0.80 / 0.57 | 0.96 / 0.23 | 1.00 / 0.00 | 1.00 / 0.00 |
| CIFAR10 | CIFAR10-C snow 2 | 0.57 / 0.90 | 0.85 / 0.61 | 0.74 / 0.76 | 0.74 / 0.80 | 0.96 / 0.18 | 0.73 / 0.74 | 0.69 / 0.95 | 1.00 / 0.00 | 0.99 / 0.01 |
| CIFAR10 | CIFAR10-C zoom blur 2 | 0.59 / 0.89 | 0.56 / 0.96 | 0.71 / 0.77 | 0.73 / 0.81 | 0.83 / 0.56 | 0.77 / 0.66 | 1.00 / 0.00 | 1.00 / 0.00 | 1.00 / 0.00 |
| CIFAR10 | CIFAR10-C brightness 5 | 0.56 / 0.88 | 0.77 / 0.78 | 0.60 / 0.90 | 0.63 / 0.89 | 0.79 / 0.72 | 0.60 / 0.92 | 1.00 / 0.00 | 1.00 / 0.00 | 0.98 / 0.14 |
| CIFAR10 | CIFAR10-C contrast 5 | 1.00 / 0.00 | 0.98 / 0.08 | 0.92 / 0.25 | 0.96 / 0.14 | 1.00 / 0.01 | 0.99 / 0.04 | 1.00 / 0.00 | 1.00 / 0.00 | 1.00 / 0.00 |
| CIFAR10 | CIFAR10-C defocus blur 5 | 0.65 / 0.82 | 0.82 / 0.71 | 0.88 / 0.39 | 0.92 / 0.25 | 0.96 / 0.18 | 0.95 / 0.14 | 1.00 / 0.00 | 1.00 / 0.00 | 1.00 / 0.00 |
| CIFAR10 | CIFAR10-C elastic 5 | 0.59 / 0.89 | 0.82 / 0.66 | 0.80 / 0.64 | 0.78 / 0.74 | 0.84 / 0.61 | 0.75 / 0.77 | 0.92 / 0.55 | 1.00 / 0.00 | 1.00 / 0.00 |
| CIFAR10 | CIFAR10-C fog 5 | 0.89 / 0.27 | 0.93 / 0.32 | 0.79 / 0.69 | 0.80 / 0.69 | 0.98 / 0.10 | 0.78 / 0.61 | 1.00 / 0.00 | 1.00 / 0.00 | 1.00 / 0.00 |
| CIFAR10 | CIFAR10-C frost 5 | 0.70 / 0.68 | 0.94 / 0.27 | 0.84 / 0.59 | 0.84 / 0.60 | 0.98 / 0.07 | 0.79 / 0.61 | 0.99 / 0.04 | 1.00 / 0.00 | 1.00 / 0.00 |
| CIFAR10 | CIFAR10-C frosted glass blur 5 | 0.55 / 0.92 | 0.97 / 0.11 | 0.90 / 0.37 | 0.89 / 0.39 | 0.99 / 0.05 | 0.98 / 0.07 | 1.00 / 0.00 | 1.00 / 0.00 | 1.00 / 0.00 |
| CIFAR10 | CIFAR10-C gaussian noise 5 | 0.67 / 0.75 | 1.00 / 0.01 | 0.91 / 0.30 | 0.96 / 0.17 | 1.00 / 0.00 | 1.00 / 0.00 | 1.00 / 0.00 | 1.00 / 0.00 | 1.00 / 0.00 |
| CIFAR10 | CIFAR10-C impulse noise 5 | 0.83 / 0.50 | 0.99 / 0.04 | 0.94 / 0.17 | 0.98 / 0.09 | 1.00 / 0.00 | 1.00 / 0.00 | 1.00 / 0.00 | 1.00 / 0.00 | 1.00 / 0.00 |
| CIFAR10 | CIFAR10-C jpeg compression 5 | 0.51 / 0.94 | 0.73 / 0.87 | 0.79 / 0.68 | 0.81 / 0.63 | 0.82 / 0.61 | 0.68 / 0.88 | 1.00 / 0.00 | 1.00 / 0.00 | 1.00 / 0.00 |
| CIFAR10 | CIFAR10-C motion blur 5 | 0.66 / 0.82 | 0.81 / 0.71 | 0.84 / 0.59 | 0.86 / 0.50 | 0.91 / 0.27 | 0.86 / 0.45 | 1.00 / 0.00 | 1.00 / 0.00 | 1.00 / 0.00 |
| CIFAR10 | CIFAR10-C pixelate 5 | 0.56 / 0.92 | 0.98 / 0.07 | 0.84 / 0.59 | 0.88 / 0.49 | 0.98 / 0.04 | 0.97 / 0.12 | 0.99 / 0.07 | 1.00 / 0.00 | 1.00 / 0.00 |
| CIFAR10 | CIFAR10-C shot noise 5 | 0.66 / 0.77 | 0.99 / 0.01 | 0.91 / 0.30 | 0.95 / 0.20 | 1.00 / 0.00 | 1.00 / 0.00 | 1.00 / 0.00 | 1.00 / 0.00 | 1.00 / 0.00 |
| CIFAR10 | CIFAR10-C snow 5 | 0.61 / 0.84 | 0.91 / 0.41 | 0.77 / 0.70 | 0.79 / 0.72 | 0.95 / 0.25 | 0.72 / 0.74 | 0.82 / 0.80 | 1.00 / 0.00 | 1.00 / 0.00 |
| CIFAR10 | CIFAR10-C zoom blur 5 | 0.63 / 0.84 | 0.72 / 0.90 | 0.87 / 0.48 | 0.88 / 0.43 | 0.96 / 0.12 | 0.91 / 0.30 | 1.00 / 0.00 | 1.00 / 0.00 | 1.00 / 0.00 |
| CIFAR100 | CIFAR100-C brightness 2 | 0.53 / 0.94 | 0.54 / 0.93 | 0.52 / 0.94 | 0.52 / 0.94 | 0.55 / 0.93 | 0.55 / 0.94 | 0.52 / 0.95 | 0.86 / 0.56 | 0.71 / 0.81 |
| CIFAR100 | CIFAR100-C contrast 2 | 0.94 / 0.21 | 0.62 / 0.87 | 0.60 / 0.90 | 0.58 / 0.90 | 0.75 / 0.71 | 0.66 / 0.83 | 0.60 / 0.91 | 0.96 / 0.22 | 0.92 / 0.48 |
| CIFAR100 | CIFAR100-C defocus blur 2 | 0.56 / 0.92 | 0.44 / 0.96 | 0.55 / 0.92 | 0.54 / 0.93 | 0.70 / 0.82 | 0.69 / 0.80 | 0.52 / 0.94 | 0.91 / 0.48 | 0.88 / 0.60 |
| CIFAR100 | CIFAR100-C elastic 2 | 0.55 / 0.93 | 0.32 / 0.97 | 0.61 / 0.88 | 0.59 / 0.89 | 0.68 / 0.85 | 0.66 / 0.85 | 0.56 / 0.92 | 0.93 / 0.43 | 0.87 / 0.60 |
| CIFAR100 | CIFAR100-C fog 2 | 0.82 / 0.52 | 0.59 / 0.90 | 0.55 / 0.93 | 0.53 / 0.93 | 0.75 / 0.72 | 0.73 / 0.73 | 0.54 / 0.93 | 0.93 / 0.33 | 0.85 / 0.64 |
| CIFAR100 | CIFAR100-C frost 2 | 0.53 / 0.90 | 0.73 / 0.76 | 0.71 / 0.79 | 0.65 / 0.83 | 0.82 / 0.65 | 0.71 / 0.75 | 0.66 / 0.89 | 0.98 / 0.09 | 0.99 / 0.03 |
| CIFAR100 | CIFAR100-C frosted glass blur 2 | 0.50 / 0.95 | 0.78 / 0.66 | 0.79 / 0.61 | 0.72 / 0.72 | 0.98 / 0.04 | 0.96 / 0.17 | 1.00 / 0.00 | 1.00 / 0.00 | 1.00 / 0.00 |
| CIFAR100 | CIFAR100-C gaussian noise 2 | 0.56 / 0.89 | 0.74 / 0.68 | 0.81 / 0.59 | 0.79 / 0.56 | 0.99 / 0.03 | 0.68 / 0.80 | 0.99 / 0.01 | 1.00 / 0.00 | 1.00 / 0.00 |
| CIFAR100 | CIFAR100-C impulse noise 2 | 0.61 / 0.85 | 0.83 / 0.59 | 0.82 / 0.59 | 0.83 / 0.55 | 0.99 / 0.03 | 0.69 / 0.78 | 0.99 / 0.00 | 1.00 / 0.00 | 1.00 / 0.00 |
| CIFAR100 | CIFAR100-C jpeg compression 2 | 0.51 / 0.94 | 0.63 / 0.86 | 0.70 / 0.79 | 0.69 / 0.79 | 0.71 / 0.82 | 0.66 / 0.89 | 0.70 / 0.88 | 1.00 / 0.01 | 0.98 / 0.08 |
| CIFAR100 | CIFAR100-C motion blur 2 | 0.60 / 0.89 | 0.49 / 0.92 | 0.68 / 0.82 | 0.63 / 0.85 | 0.82 / 0.57 | 0.67 / 0.79 | 0.68 / 0.89 | 1.00 / 0.00 | 0.97 / 0.17 |
| CIFAR100 | CIFAR100-C pixelate 2 | 0.52 / 0.94 | 0.76 / 0.75 | 0.63 / 0.88 | 0.61 / 0.87 | 0.87 / 0.49 | 0.57 / 0.91 | 0.77 / 0.89 | 1.00 / 0.00 | 0.98 / 0.11 |
| CIFAR100 | CIFAR100-C shot noise 2 | 0.53 / 0.92 | 0.70 / 0.77 | 0.75 / 0.73 | 0.73 / 0.72 | 0.94 / 0.23 | 0.60 / 0.85 | 0.83 / 0.79 | 1.00 / 0.00 | 1.00 / 0.00 |
| CIFAR100 | CIFAR100-C snow 2 | 0.55 / 0.91 | 0.76 / 0.73 | 0.73 / 0.77 | 0.69 / 0.80 | 0.93 / 0.36 | 0.61 / 0.88 | 0.79 / 0.81 | 0.99 / 0.02 | 0.99 / 0.05 |
| CIFAR100 | CIFAR100-C zoom blur 2 | 0.59 / 0.90 | 0.41 / 0.94 | 0.69 / 0.81 | 0.63 / 0.85 | 0.83 / 0.46 | 0.68 / 0.81 | 0.71 / 0.90 | 1.00 / 0.00 | 0.99 / 0.06 |
| CIFAR100 | CIFAR100-C brightness 5 | 0.53 / 0.93 | 0.70 / 0.83 | 0.64 / 0.86 | 0.62 / 0.87 | 0.74 / 0.76 | 0.72 / 0.77 | 0.60 / 0.90 | 0.98 / 0.11 | 0.96 / 0.29 |
| CIFAR100 | CIFAR100-C contrast 5 | 1.00 / 0.01 | 0.79 / 0.55 | 0.78 / 0.57 | 0.82 / 0.53 | 0.99 / 0.03 | 0.98 / 0.12 | 0.99 / 0.00 | 1.00 / 0.00 | 1.00 / 0.00 |
| CIFAR100 | CIFAR100-C defocus blur 5 | 0.65 / 0.84 | 0.54 / 0.83 | 0.81 / 0.63 | 0.75 / 0.67 | 0.96 / 0.15 | 0.95 / 0.19 | 0.99 / 0.01 | 1.00 / 0.00 | 1.00 / 0.00 |
| CIFAR100 | CIFAR100-C elastic 5 | 0.59 / 0.90 | 0.73 / 0.78 | 0.72 / 0.76 | 0.69 / 0.79 | 0.71 / 0.81 | 0.71 / 0.82 | 0.63 / 0.85 | 1.00 / 0.00 | 0.99 / 0.07 |
| CIFAR100 | CIFAR100-C fog 5 | 0.90 / 0.25 | 0.80 / 0.60 | 0.75 / 0.70 | 0.70 / 0.73 | 0.92 / 0.40 | 0.80 / 0.67 | 0.95 / 0.46 | 1.00 / 0.00 | 0.99 / 0.01 |
| CIFAR100 | CIFAR100-C frost 5 | 0.73 / 0.68 | 0.78 / 0.63 | 0.79 / 0.63 | 0.72 / 0.71 | 0.94 / 0.26 | 0.66 / 0.77 | 0.95 / 0.45 | 1.00 / 0.00 | 1.00 / 0.00 |
| CIFAR100 | CIFAR100-C frosted glass blur 5 | 0.55 / 0.93 | 0.80 / 0.63 | 0.80 / 0.60 | 0.73 / 0.72 | 0.97 / 0.08 | 0.76 / 0.70 | 1.00 / 0.00 | 1.00 / 0.00 | 1.00 / 0.00 |
| CIFAR100 | CIFAR100-C gaussian noise 5 | 0.67 / 0.77 | 0.81 / 0.54 | 0.84 / 0.42 | 0.85 / 0.41 | 1.00 / 0.00 | 1.00 / 0.00 | 1.00 / 0.00 | 1.00 / 0.00 | 1.00 / 0.00 |
| CIFAR100 | CIFAR100-C impulse noise 5 | 0.84 / 0.48 | 0.88 / 0.37 | 0.86 / 0.42 | 0.96 / 0.20 | 1.00 / 0.00 | 1.00 / 0.00 | 1.00 / 0.00 | 1.00 / 0.00 | 1.00 / 0.00 |
| CIFAR100 | CIFAR100-C jpeg compression 5 | 0.51 / 0.94 | 0.69 / 0.79 | 0.77 / 0.71 | 0.75 / 0.69 | 0.82 / 0.61 | 0.73 / 0.82 | 0.94 / 0.50 | 1.00 / 0.00 | 0.99 / 0.04 |
| CIFAR100 | CIFAR100-C motion blur 5 | 0.66 / 0.83 | 0.56 / 0.86 | 0.76 / 0.70 | 0.71 / 0.74 | 0.91 / 0.29 | 0.77 / 0.64 | 0.87 / 0.71 | 1.00 / 0.01 | 0.99 / 0.03 |
| CIFAR100 | CIFAR100-C pixelate 5 | 0.55 / 0.92 | 0.95 / 0.17 | 0.80 / 0.59 | 0.80 / 0.58 | 0.99 / 0.01 | 0.98 / 0.08 | 0.99 / 0.00 | 1.00 / 0.00 | 1.00 / 0.00 |
| CIFAR100 | CIFAR100-C shot noise 5 | 0.66 / 0.79 | 0.84 / 0.50 | 0.83 / 0.47 | 0.84 / 0.44 | 1.00 / 0.00 | 1.00 / 0.01 | 0.99 / 0.00 | 1.00 / 0.00 | 1.00 / 0.00 |
| CIFAR100 | CIFAR100-C snow 5 | 0.56 / 0.90 | 0.80 / 0.63 | 0.77 / 0.70 | 0.73 / 0.73 | 0.90 / 0.45 | 0.63 / 0.87 | 0.79 / 0.77 | 0.99 / 0.06 | 0.99 / 0.02 |
| CIFAR100 | CIFAR100-C zoom blur 5 | 0.64 / 0.85 | 0.49 / 0.88 | 0.79 / 0.66 | 0.74 / 0.70 | 0.92 / 0.29 | 0.93 / 0.24 | 0.92 / 0.62 | 1.00 / 0.00 | 1.00 / 0.00 |
| Tiny ImageNet | Tiny ObjectNet | 0.51 / 0.96 | 0.70 / 0.68 | 0.82 / 0.51 | 0.79 / 0.63 | 0.75 / 0.74 | 0.72 / 0.75 | 0.99 / 0.02 | 0.99 / 0.02 | 0.98 / 0.12 |
| | Average | 0.63 / 0.79 | 0.76 / 0.60 | 0.76 / 0.64 | 0.76 / 0.64 | 0.87 / 0.38 | 0.79 / 0.54 | 0.86 / 0.40 | 0.98 / 0.08 | 0.96 / 0.15 |

## G.2 RESULTS WITH A SMALLER TEST SET

Table 6: Experiments with a test set of size 1000, with an equal number of ID and OOD test samples. For RETO and vanilla ensembles, we train 5 ResNet20 models for each setting.

| ID data | OOD data | kNN | DPN | Vanilla Ensembles | OE | Mahal. | Mahal-T | MCD | RETO (pretrained) |
|---|---|---|---|---|---|---|---|---|---|
| | | | | | AUROC ↑ / FPR@95 ↓ | | | | |
| SVHN | CIFAR10 | 0.92 / 0.32 | 1.00 / 0.00 | 0.97 / 0.12 | 1.00 / 0.00 | 0.99 / 0.02 | 0.99 / 0.05 | 0.97 / 0.15 | 1.00 / 0.01 |
| CIFAR10 | SVHN | 0.81 / 0.74 | 0.95 / 0.15 | 0.92 / 0.22 | 0.97 / 0.11 | 0.99 / 0.04 | 0.99 / 0.04 | 1.00 / 0.02 | 1.00 / 0.00 |
| CIFAR100 | SVHN | 0.83 / 0.71 | 0.77 / 0.56 | 0.84 / 0.52 | 0.82 / 0.50 | 0.98 / 0.10 | 0.98 / 0.08 | 0.97 / 0.27 | 0.99 / 0.00 |
| SVHN[0:4] | SVHN[5:9] | 0.54 / 0.92 | 0.87 / 0.81 | 0.92 / 0.31 | 0.85 / 0.48 | 0.92 / 0.29 | 0.91 / 0.37 | 0.91 / 0.49 | 0.97 / 0.14 |
| CIFAR10[0:4] | CIFAR10[5:9] | 0.59 / 0.86 | 0.82 / 0.68 | 0.80 / 0.61 | 0.82 / 0.59 | 0.79 / 0.73 | 0.64 / 0.87 | 0.69 / 0.75 | 0.87 / 0.50 |
| CIFAR100[0:49] | CIFAR100[50:99] | 0.51 / 0.96 | 0.70 / 0.74 | 0.78 / 0.65 | 0.74 / 0.69 | 0.72 / 0.80 | 0.72 / 0.81 | 0.70 / 0.74 | 0.79 / 0.62 |
| CIFAR10 | CIFAR10-C brightness 2 | 0.54 / 0.93 | 0.55 / 0.93 | 0.51 / 0.95 | 0.52 / 0.94 | 0.58 / 0.92 | 0.55 / 0.93 | 0.52 / 0.94 | 0.57 / 0.91 |
| CIFAR10 | CIFAR10-C contrast 2 | 0.94 / 0.20 | 0.69 / 0.86 | 0.59 / 0.91 | 0.62 / 0.89 | 0.82 / 0.56 | 0.69 / 0.79 | 0.82 / 0.91 | 0.91 / 0.39 |
| CIFAR10 | CIFAR10-C defocus blur 2 | 0.56 / 0.91 | 0.48 / 0.95 | 0.54 / 0.93 | 0.56 / 0.93 | 0.67 / 0.81 | 0.58 / 0.90 | 0.60 / 0.90 | 0.80 / 0.66 |
| CIFAR10 | CIFAR10-C elastic 2 | 0.56 / 0.92 | 0.47 / 0.97 | 0.63 / 0.87 | 0.65 / 0.87 | 0.71 / 0.76 | 0.66 / 0.85 | 0.68 / 0.91 | 0.84 / 0.62 |
| CIFAR10 | CIFAR10-C fog 2 | 0.81 / 0.52 | 0.61 / 0.89 | 0.53 / 0.93 | 0.55 / 0.93 | 0.81 / 0.62 | 0.71 / 0.73 | 0.58 / 0.92 | 0.71 / 0.77 |
| CIFAR10 | CIFAR10-C frost 2 | 0.50 / 0.92 | 0.84 / 0.60 | 0.69 / 0.84 | 0.69 / 0.85 | 0.90 / 0.42 | 0.72 / 0.73 | 0.59 / 0.95 | 0.98 / 0.12 |
| CIFAR10 | CIFAR10-C frosted glass blur 2 | 0.51 / 0.95 | 0.98 / 0.09 | 0.89 / 0.42 | 0.86 / 0.50 | 0.99 / 0.02 | 0.99 / 0.04 | 0.99 / 0.04 | 1.00 / 0.00 |
| CIFAR10 | CIFAR10-C gaussian noise 2 | 0.57 / 0.89 | 0.97 / 0.13 | 0.83 / 0.62 | 0.85 / 0.58 | 1.00 / 0.01 | 0.99 / 0.02 | 1.00 / 0.00 | 1.00 / 0.00 |
| CIFAR10 | CIFAR10-C impulse noise 2 | 0.61 / 0.85 | 0.95 / 0.23 | 0.80 / 0.68 | 0.81 / 0.61 | 1.00 / 0.02 | 0.99 / 0.04 | 1.00 / 0.00 | 1.00 / 0.00 |
| CIFAR10 | CIFAR10-C jpeg compression 2 | 0.51 / 0.95 | 0.65 / 0.92 | 0.71 / 0.81 | 0.73 / 0.77 | 0.75 / 0.74 | 0.68 / 0.84 | 1.00 / 0.00 | 0.98 / 0.14 |
| CIFAR10 | CIFAR10-C motion blur 2 | 0.60 / 0.88 | 0.65 / 0.88 | 0.71 / 0.81 | 0.74 / 0.77 | 0.80 / 0.69 | 0.79 / 0.61 | 0.99 / 0.01 | 0.97 / 0.23 |
| CIFAR10 | CIFAR10-C pixelate 2 | 0.52 / 0.94 | 0.82 / 0.67 | 0.65 / 0.88 | 0.66 / 0.85 | 0.79 / 0.71 | 0.66 / 0.86 | 0.93 / 0.73 | 0.96 / 0.25 |
| CIFAR10 | CIFAR10-C shot noise 2 | 0.53 / 0.92 | 0.87 / 0.61 | 0.74 / 0.79 | 0.75 / 0.82 | 0.98 / 0.06 | 0.80 / 0.57 | 0.96 / 0.23 | 1.00 / 0.06 |
| CIFAR10 | CIFAR10-C snow 2 | 0.57 / 0.90 | 0.85 / 0.61 | 0.74 / 0.76 | 0.74 / 0.80 | 0.96 / 0.18 | 0.73 / 0.74 | 0.69 / 0.95 | 0.99 / 0.11 |
| CIFAR10 | CIFAR10-C zoom blur 2 | 0.59 / 0.89 | 0.56 / 0.96 | 0.71 / 0.77 | 0.73 / 0.81 | 0.83 / 0.56 | 0.77 / 0.66 | 1.00 / 0.00 | 0.99 / 0.06 |
| CIFAR10 | CIFAR10-C brightness 5 | 0.56 / 0.88 | 0.77 / 0.78 | 0.60 / 0.90 | 0.63 / 0.89 | 0.78 / 0.73 | 0.79 / 0.72 | 0.60 / 0.92 | 0.92 / 0.33 |
| CIFAR10 | CIFAR10-C contrast 5 | 1.00 / 0.00 | 0.98 / 0.08 | 0.92 / 0.25 | 0.96 / 0.14 | 1.00 / 0.01 | 0.99 / 0.04 | 1.00 / 0.00 | 1.00 / 0.00 |
| CIFAR10 | CIFAR10-C defocus blur 5 | 0.65 / 0.82 | 0.82 / 0.71 | 0.88 / 0.39 | 0.92 / 0.25 | 0.96 / 0.18 | 0.95 / 0.14 | 1.00 / 0.00 | 1.00 / 0.00 |
| CIFAR10 | CIFAR10-C elastic 5 | 0.59 / 0.89 | 0.82 / 0.66 | 0.80 / 0.64 | 0.78 / 0.74 | 0.84 / 0.61 | 0.75 / 0.77 | 0.92 / 0.55 | 0.98 / 0.15 |
| CIFAR10 | CIFAR10-C fog 5 | 0.89 / 0.27 | 0.93 / 0.32 | 0.79 / 0.69 | 0.80 / 0.69 | 0.98 / 0.04 | 0.78 / 0.61 | 1.00 / 0.00 | 0.99 / 0.02 |
| CIFAR10 | CIFAR10-C frost 5 | 0.70 / 0.68 | 0.94 / 0.27 | 0.84 / 0.59 | 0.84 / 0.60 | 0.98 / 0.07 | 0.79 / 0.61 | 0.99 / 0.04 | 1.00 / 0.00 |
| CIFAR10 | CIFAR10-C frosted glass blur 5 | 0.55 / 0.92 | 0.97 / 0.11 | 0.90 / 0.37 | 0.89 / 0.39 | 0.99 / 0.05 | 0.98 / 0.07 | 1.00 / 0.00 | 1.00 / 0.00 |
| CIFAR10 | CIFAR10-C gaussian noise 5 | 0.67 / 0.75 | 1.00 / 0.01 | 0.91 / 0.30 | 0.96 / 0.17 | 1.00 / 0.00 | 1.00 / 0.00 | 1.00 / 0.00 | 1.00 / 0.00 |
| CIFAR10 | CIFAR10-C impulse noise 5 | 0.83 / 0.50 | 0.99 / 0.04 | 0.94 / 0.17 | 0.98 / 0.09 | 1.00 / 0.00 | 1.00 / 0.00 | 1.00 / 0.00 | 1.00 / 0.00 |
| CIFAR10 | CIFAR10-C jpeg compression 5 | 0.51 / 0.94 | 0.73 / 0.87 | 0.79 / 0.68 | 0.81 / 0.63 | 0.82 / 0.61 | 0.68 / 0.88 | 1.00 / 0.00 | 1.00 / 0.02 |
| CIFAR10 | CIFAR10-C motion blur 5 | 0.66 / 0.82 | 0.81 / 0.71 | 0.84 / 0.59 | 0.86 / 0.50 | 0.91 / 0.27 | 0.86 / 0.45 | 1.00 / 0.00 | 0.99 / 0.03 |
| CIFAR10 | CIFAR10-C pixelate 5 | 0.56 / 0.92 | 0.98 / 0.07 | 0.84 / 0.59 | 0.88 / 0.49 | 0.98 / 0.04 | 0.97 / 0.12 | 0.99 / 0.07 | 0.99 / 0.00 |
| CIFAR10 | CIFAR10-C shot noise 5 | 0.66 / 0.77 | 0.99 / 0.01 | 0.91 / 0.30 | 0.95 / 0.20 | 1.00 / 0.00 | 1.00 / 0.01 | 1.00 / 0.00 | 1.00 / 0.00 |
| CIFAR10 | CIFAR10-C snow 5 | 0.61 / 0.84 | 0.91 / 0.41 | 0.77 / 0.70 | 0.79 / 0.72 | 0.95 / 0.25 | 0.72 / 0.74 | 0.82 / 0.80 | 0.98 / 0.14 |
| CIFAR10 | CIFAR10-C zoom blur 5 | 0.63 / 0.84 | 0.72 / 0.90 | 0.87 / 0.48 | 0.88 / 0.43 | 0.96 / 0.12 | 0.91 / 0.30 | 1.00 / 0.00 | 1.00 / 0.00 |
| CIFAR100 | CIFAR100-C brightness 2 | 0.53 / 0.94 | 0.54 / 0.93 | 0.52 / 0.94 | 0.52 / 0.94 | 0.55 / 0.93 | 0.55 / 0.94 | 0.52 / 0.95 | 0.55 / 0.93 |
| CIFAR100 | CIFAR100-C contrast 2 | 0.94 / 0.21 | 0.62 / 0.87 | 0.60 / 0.90 | 0.58 / 0.90 | 0.75 / 0.71 | 0.66 / 0.83 | 0.60 / 0.91 | 0.73 / 0.83 |
| CIFAR100 | CIFAR100-C defocus blur 2 | 0.56 / 0.92 | 0.44 / 0.96 | 0.55 / 0.92 | 0.54 / 0.93 | 0.70 / 0.82 | 0.69 / 0.80 | 0.52 / 0.94 | 0.63 / 0.87 |
| CIFAR100 | CIFAR100-C elastic 2 | 0.55 / 0.93 | 0.32 / 0.97 | 0.61 / 0.88 | 0.59 / 0.89 | 0.68 / 0.86 | 0.66 / 0.85 | 0.56 / 0.92 | 0.68 / 0.85 |
| CIFAR100 | CIFAR100-C fog 2 | 0.82 / 0.52 | 0.59 / 0.90 | 0.55 / 0.93 | 0.53 / 0.93 | 0.75 / 0.72 | 0.73 / 0.73 | 0.54 / 0.93 | 0.61 / 0.91 |
| CIFAR100 | CIFAR100-C frost 2 | 0.53 / 0.90 | 0.73 / 0.76 | 0.71 / 0.79 | 0.65 / 0.83 | 0.82 / 0.65 | 0.71 / 0.75 | 0.66 / 0.89 | 0.94 / 0.50 |
| CIFAR100 | CIFAR100-C frosted glass blur 2 | 0.50 / 0.95 | 0.78 / 0.66 | 0.79 / 0.61 | 0.72 / 0.72 | 0.98 / 0.04 | 0.96 / 0.17 | 1.00 / 0.00 | 1.00 / 0.00 |
| CIFAR100 | CIFAR100-C gaussian noise 2 | 0.56 / 0.89 | 0.74 / 0.68 | 0.81 / 0.59 | 0.79 / 0.56 | 0.99 / 0.03 | 0.68 / 0.80 | 0.99 / 0.01 | 0.99 / 0.00 |
| CIFAR100 | CIFAR100-C impulse noise 2 | 0.61 / 0.85 | 0.83 / 0.59 | 0.82 / 0.59 | 0.83 / 0.55 | 0.99 / 0.03 | 0.69 / 0.78 | 0.99 / 0.00 | 0.99 / 0.00 |
| CIFAR100 | CIFAR100-C jpeg compression 2 | 0.51 / 0.94 | 0.63 / 0.86 | 0.70 / 0.79 | 0.69 / 0.79 | 0.71 / 0.72 | 0.66 / 0.89 | 0.70 / 0.88 | 0.88 / 0.61 |
| CIFAR100 | CIFAR100-C motion blur 2 | 0.60 / 0.89 | 0.49 / 0.92 | 0.68 / 0.82 | 0.63 / 0.85 | 0.82 / 0.57 | 0.67 / 0.79 | 0.68 / 0.89 | 0.86 / 0.62 |
| CIFAR100 | CIFAR100-C pixelate 2 | 0.52 / 0.94 | 0.76 / 0.75 | 0.63 / 0.88 | 0.61 / 0.87 | 0.87 / 0.49 | 0.57 / 0.91 | 0.77 / 0.89 | 0.86 / 0.60 |
| CIFAR100 | CIFAR100-C shot noise 2 | 0.53 / 0.92 | 0.70 / 0.77 | 0.75 / 0.73 | 0.73 / 0.72 | 0.94 / 0.23 | 0.60 / 0.85 | 0.83 / 0.79 | 0.98 / 0.11 |
| CIFAR100 | CIFAR100-C snow 2 | 0.55 / 0.91 | 0.76 / 0.73 | 0.73 / 0.77 | 0.69 / 0.80 | 0.93 / 0.36 | 0.61 / 0.88 | 0.79 / 0.81 | 0.94 / 0.42 |
| CIFAR100 | CIFAR100-C zoom blur 2 | 0.59 / 0.90 | 0.41 / 0.94 | 0.69 / 0.81 | 0.63 / 0.85 | 0.83 / 0.46 | 0.68 / 0.81 | 0.71 / 0.90 | 0.91 / 0.59 |
| CIFAR100 | CIFAR100-C brightness 5 | 0.53 / 0.93 | 0.70 / 0.83 | 0.64 / 0.86 | 0.62 / 0.87 | 0.74 / 0.76 | 0.72 / 0.77 | 0.60 / 0.90 | 0.81 / 0.75 |
| CIFAR100 | CIFAR100-C contrast 5 | 1.00 / 0.01 | 0.79 / 0.55 | 0.78 / 0.57 | 0.82 / 0.53 | 0.99 / 0.03 | 0.98 / 0.12 | 0.99 / 0.00 | 0.99 / 0.01 |
| CIFAR100 | CIFAR100-C defocus blur 5 | 0.65 / 0.84 | 0.54 / 0.83 | 0.81 / 0.63 | 0.75 / 0.67 | 0.96 / 0.15 | 0.95 / 0.19 | 0.99 / 0.01 | 0.99 / 0.03 |
| CIFAR100 | CIFAR100-C elastic 5 | 0.59 / 0.90 | 0.73 / 0.78 | 0.72 / 0.76 | 0.69 / 0.79 | 0.71 / 0.81 | 0.71 / 0.82 | 0.63 / 0.85 | 0.89 / 0.57 |
| CIFAR100 | CIFAR100-C fog 5 | 0.90 / 0.25 | 0.80 / 0.60 | 0.75 / 0.70 | 0.70 / 0.73 | 0.92 / 0.40 | 0.80 / 0.67 | 0.95 / 0.46 | 0.96 / 0.34 |
| CIFAR100 | CIFAR100-C frost 5 | 0.73 / 0.68 | 0.78 / 0.63 | 0.79 / 0.63 | 0.72 / 0.71 | 0.94 / 0.26 | 0.66 / 0.77 | 0.95 / 0.45 | 0.98 / 0.06 |
| CIFAR100 | CIFAR100-C frosted glass blur 5 | 0.55 / 0.93 | 0.80 / 0.63 | 0.84 / 0.60 | 0.73 / 0.72 | 0.97 / 0.08 | 0.76 / 0.70 | 1.00 / 0.00 | 1.00 / 0.00 |
| CIFAR100 | CIFAR100-C gaussian noise 5 | 0.67 / 0.77 | 0.81 / 0.54 | 0.84 / 0.42 | 0.85 / 0.41 | 1.00 / 0.00 | 1.00 / 0.00 | 1.00 / 0.00 | 1.00 / 0.00 |
| CIFAR100 | CIFAR100-C impulse noise 5 | 0.84 / 0.48 | 0.88 / 0.37 | 0.86 / 0.42 | 0.96 / 0.20 | 1.00 / 0.00 | 1.00 / 0.00 | 1.00 / 0.00 | 1.00 / 0.00 |
| CIFAR100 | CIFAR100-C jpeg compression 5 | 0.51 / 0.94 | 0.69 / 0.79 | 0.77 / 0.71 | 0.75 / 0.69 | 0.82 / 0.61 | 0.73 / 0.82 | 0.94 / 0.50 | 0.95 / 0.39 |
| CIFAR100 | CIFAR100-C motion blur 5 | 0.66 / 0.83 | 0.56 / 0.86 | 0.76 / 0.70 | 0.71 / 0.74 | 0.91 / 0.29 | 0.77 / 0.64 | 0.87 / 0.71 | 0.95 / 0.40 |
| CIFAR100 | CIFAR100-C pixelate 5 | 0.55 / 0.92 | 0.95 / 0.17 | 0.80 / 0.59 | 0.80 / 0.58 | 0.99 / 0.01 | 0.98 / 0.08 | 0.99 / 0.00 | 1.00 / 0.00 |
| CIFAR100 | CIFAR100-C shot noise 5 | 0.66 / 0.79 | 0.84 / 0.50 | 0.83 / 0.47 | 0.84 / 0.44 | 1.00 / 0.00 | 1.00 / 0.01 | 0.99 / 0.00 | 1.00 / 0.00 |
| CIFAR100 | CIFAR100-C snow 5 | 0.56 / 0.90 | 0.80 / 0.63 | 0.77 / 0.70 | 0.73 / 0.73 | 0.90 / 0.45 | 0.63 / 0.87 | 0.79 / 0.77 | 0.96 / 0.37 |
| CIFAR100 | CIFAR100-C zoom blur 5 | 0.64 / 0.85 | 0.49 / 0.88 | 0.79 / 0.66 | 0.74 / 0.70 | 0.92 / 0.29 | 0.93 / 0.24 | 0.92 / 0.62 | 0.98 / 0.06 |
| | Average | 0.64 / 0.79 | 0.76 / 0.60 | 0.76 / 0.64 | 0.76 / 0.64 | 0.88 / 0.37 | 0.79 / 0.53 | 0.85 / 0.41 | 0.91 / 0.27 |

### G.3 RESULTS WITH VGG

Table 7: Results on all setting where we used ensembles of 5 VGG16 models for RETO and vanilla. For the corruption data sets, we report for each metric the average taken over all corruptions (A), and the value for the worst-case setting (W).

| ID data | OOD data | kNN | DPN | Vanilla Ensembles | OE | Mahal. | Mahal-T | MCD | RETO (rand init) | RETO (pretrained) |
|---|---|---|---|---|---|---|---|---|---|---|
| | | | | | AUROC ↑ / FPR@95 ↓ | | | | | |
| SVHN | CIFAR10 | 0.92 / 0.32 | *1.00 / 0.00* | 0.97 / 0.12 | *1.00 / 0.00* | 0.99 / 0.02 | 0.99 / 0.05 | 0.97 / 0.15 | **0.99** / **0.02** | **0.99** / 0.06 |
| CIFAR10 | SVHN | 0.81 / 0.74 | 0.95 / 0.15 | 0.88 / 0.31 | 0.97 / 0.11 | 0.99 / 0.04 | 0.99 / 0.04 | *1.00 / 0.02* | **1.00** / **0.00** | **1.00** / **0.00** |
| SVHN[0:4] | SVHN[5:9] | 0.54 / 0.92 | 0.87 / 0.81 | 0.89 / 0.40 | 0.85 / 0.48 | *0.92 / 0.29* | 0.91 / 0.37 | 0.91 / 0.49 | **0.94** / **0.34** | 0.93 / 0.37 |
| CIFAR10[0:4] | CIFAR10[5:9] | 0.59 / 0.86 | *0.82* / 0.68 | 0.74 / 0.71 | *0.82 / 0.59* | 0.79 / 0.73 | 0.64 / 0.87 | 0.69 / 0.75 | **0.95** / **0.26** | 0.91 / 0.37 |
| CIFAR10 | CIFAR10-C sev 2 (A) | 0.59 / 0.84 | 0.73 / 0.69 | 0.66 / 0.83 | 0.70 / 0.80 | *0.84 / 0.47* | 0.75 / 0.62 | 0.82 / 0.50 | **0.98** / **0.07** | 0.94 / 0.21 |
| CIFAR10 | CIFAR10-C sev 2 (W) | 0.50 / *0.92* | 0.47 / 0.97 | 0.51 / 0.95 | 0.52 / 0.94 | *0.58 / 0.92* | 0.55 / 0.93 | 0.52 / 0.94 | **0.77** / **0.67** | 0.68 / 0.81 |
| CIFAR10 | CIFAR10-C sev 5 (A) | 0.67 / 0.72 | 0.89 / 0.40 | 0.80 / 0.59 | 0.86 / 0.46 | 0.94 / 0.20 | 0.88 / 0.37 | *0.95 / 0.16* | **1.00** / **0.00** | 0.99 / 0.04 |
| CIFAR10 | CIFAR10-C sev 5 (W) | 0.51 / 0.94 | 0.72 / 0.90 | 0.58 / 0.90 | 0.63 / 0.89 | *0.78 / 0.73* | 0.68 / 0.88 | 0.60 / 0.92 | **0.99** / **0.01** | 0.95 / 0.28 |
| | Average | 0.69 / 0.73 | 0.88 / 0.45 | 0.82 / 0.49 | 0.87 / 0.41 | *0.91 / 0.29* | 0.86 / 0.39 | 0.89 / 0.35 | **0.98** / **0.11** | 0.96 / 0.17 |

### G.4 RESULTS ON MNIST AND FASHIONMNIST

Table 8: Results on MNIST/FashionMNIST settings. For RETO and vanilla ensembles, we train 5 3-hidden layer MLP models for each setting.

| ID data | OOD data | kNN | DPN | Vanilla Ensembles | OE | Mahal. | Mahal-T | MCD | RETO (rand init) | RETO (pretrained) |
|---|---|---|---|---|---|---|---|---|---|---|
| | | | | | AUROC ↑ / FPR@95 ↓ | | | | | |
| MNIST | FashionMNIST | 0.98 / 0.10 | *1.00 / 0.00* | 0.81 / 0.99 | *1.00 / 0.00* | *1.00 / 0.00* | *1.00 / 0.00* | 1.00 / 0.02 | **1.00** / **0.00** | **1.00** / **0.00** |
| FashionMNIST | MNIST | 0.98 / 0.09 | *1.00 / 0.00* | 0.87 / 0.58 | 0.68 / 0.84 | 0.99 / 0.03 | 0.99 / 0.04 | *1.00 / 0.00* | **1.00** / **0.00** | **1.00** / **0.00** |
| MNIST[0:4] | MNIST[5:9] | 0.90 / 0.42 | *0.99* / 0.03 | 0.94 / 0.28 | 0.95 / 0.22 | *0.99 / 0.02* | *0.99 / 0.02* | 0.96 / 0.24 | **1.00** / 0.02 | 0.99 / **0.02** |
| FashionMNIST[0,2,3,7,8] | FashionMNIST[1,4,5,6,9] | 0.74 / 0.67 | 0.77 / 0.85 | 0.64 / 0.93 | 0.66 / 0.88 | 0.77 / 0.80 | *0.82 / 0.61* | 0.78 / 0.70 | **0.95** / **0.27** | 0.93 / 0.38 |
| | Average | 0.90 / 0.32 | 0.94 / 0.22 | 0.82 / 0.70 | 0.82 / 0.49 | 0.94 / 0.21 | *0.95 / 0.17* | 0.94 / 0.24 | **0.99** / **0.07** | 0.98 / 0.10 |

For FashionMNIST we chose this particular split (i.e. classes 0,2,3,7,8 vs classes 1,4,5,6,9) because the two partitions are more similar to each other. This makes OOD detection more difficult than the 0-4 vs 5-9 split.

### G.5 MORE RESULTS FOR OUTLIER EXPOSURE

Table 9: Results for Outlier Exposure, when using the same corruption type, but with a higher/lower severity, as OOD data seen during training.

| ID data | OOD data | OE (trained on sev5) | OE (trained on sev2) |
|---|---|---|---|
| | | AUROC ↑ | |
| CIFAR10 | CIFAR10-C sev 2 (A) | 0.89 | N/A |
| CIFAR10 | CIFAR10-C sev 2 (W) | 0.65 | N/A |
| CIFAR10 | CIFAR10-C sev 5 (A) | N/A | 0.98 |
| CIFAR10 | CIFAR10-C sev 5 (W) | N/A | 0.78 |
| CIFAR100 | CIFAR100-C sev 2 (A) | 0.85 | N/A |
| CIFAR100 | CIFAR100-C sev 2 (W) | 0.59 | N/A |
| CIFAR100 | CIFAR100-C sev 5 (A) | N/A | 0.97 |
| CIFAR100 | CIFAR100-C sev 5 (W) | N/A | 0.67 |
| | Average | 0.87 | 0.98 |

The Outlier Exposure method needs access to a set of OOD samples during training. The numbers we report in the rest of paper for Outlier Exposure are obtained by using the TinyImages data set as the OOD samples that are seen during training. In this section we explore the use of an $OOD_{train}$ data set that is more similar to the OOD data observed at test time. This is a much easier setting for

the Outlier Exposure method: the closer $OOD_{train}$ is to $OOD_{test}$, the easier it will be for the model tuned on $OOD_{train}$ to detect the test OOD samples.

In the table below we focus only on the settings with corruptions. For each corruption type, we use the lower severity corruption as $OOD_{train}$ and evaluate on the higher severity data and vice versa. We report for each metric the average taken over all corruptions (A), and the value for the worst-case setting (W).

## H EXPERIMENTS ON CIFAR10v2

Here we will present our results on distinguishing between CIFAR10 (Krizhevsky (2009)) and CIFAR10v2 (Recht et al. (2018)), a dataset meant to be drawn from the same distribution as CIFAR10 (generated from the Tiny Images collection Torralba et al. (2008)). Recht et al. (2018) and Recht et al. (2019) argue that classifiers originally trained on CIFAR10 have a statistically significant drop in accuracy when evaluated on CIFAR10v2. Furthermore, Recht et al. (2019) argue that CIFAR10 and CIFAR10v2 come from the same distribution by training a binary classifier to distinguish between them, and observing that the accuracy is obtained is very close to random (50.1% for the randomly initialized models, and 52.9% for models with pre-trained weights).

Our experiments show that the two datasets are actually distinguishable, contrary to what previous work has argued. First, our own binary classifier trained on CIFAR10 vs CIFAR10v2 obtains a test accuracy of 67%, without any hyperparameter tuning (the model is a ResNet20 trained for 200 epochs using SGD with momentum 0.9. The learning rate is decayed by 0.2 at epochs 90, 140, 160 and 180. We use 1600 examples from each dataset for training, and we validate using 400 examples from each dataset).

Table 10: OOD detection performance on CIFAR10 vs CIFAR10v2

| ID data | OOD data | kNN | DPN | Vanilla Ensembles | OE | Mahal. | Mahal-T | MCD | RETO (rand init) | RETO (pretrained) |
|---------|----------|-----|-----|-------------------|-----|--------|---------|-----|------------------|-------------------|
| | | | | | | AUROC ↑ / FPR@95 ↓ | | | | |
| CIFAR10 | CIFAR10v2 | 0.53 / 0.93 | 0.63 / 0.91 | *0.64* / *0.87* | *0.64* / 0.88 | 0.55 / 0.92 | 0.56 / 0.93 | 0.58 / 0.90 | **0.91** / **0.20** | 0.76 / 0.74 |

Furthermore, our OOD experiments (as shown in Table 10) show that most baselines are able to distinguish between the two datasets, with RETO achieving the highest performance. The methods which require OOD data for tuning (Outlier Exposure and DPN) use CIFAR100 for tuning.

# I   DEPENDENCE ON THE TEST SET CONFIGURATION

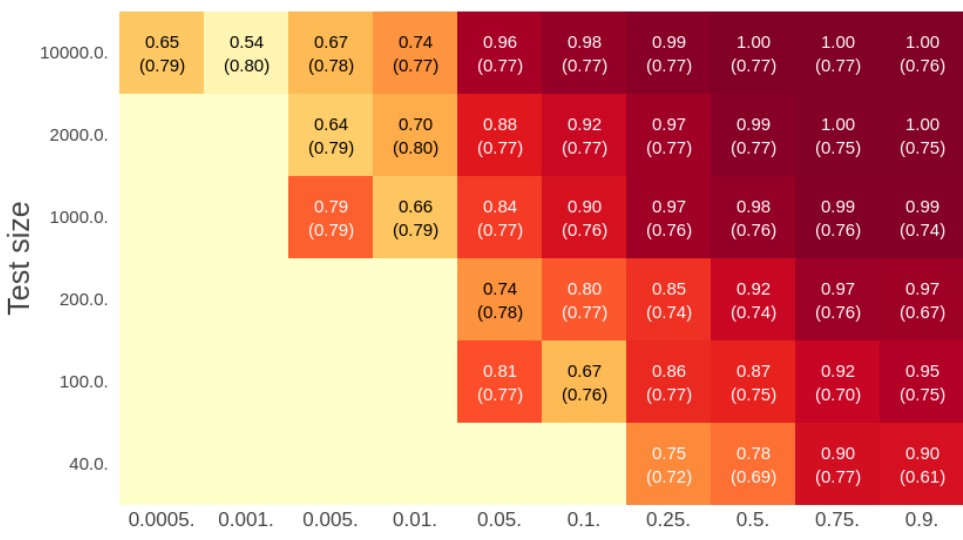

(a) Initialization from pretrained weights.

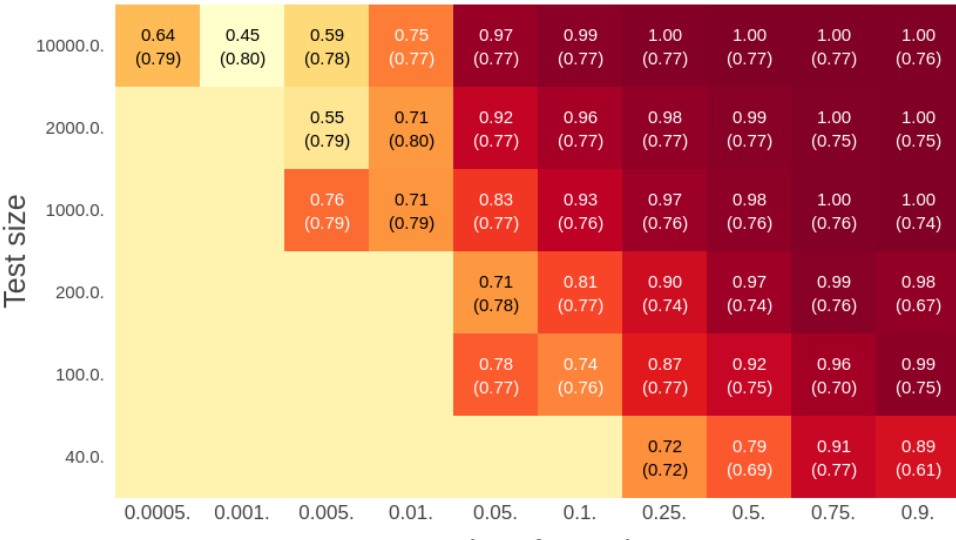

(b) Initialization from random weights.

Figure 11: AUROCs obtained with an ensemble of ResNet20 models as the composition of the test set is changed. Only settings with at least 5 OOD samples have been considered. The ID samples are from CIFAR10, while the OOD samples are from CIFAR10-C/snow5. For comparison, we provide in parentheses AUROC values for a vanilla ensemble trained once on the training set and evaluated on each test set configuration. In 11a, the models are initialized from pretrained weights and fine tuned. In 11b, the models are initialized with random weights and trained from scratch.

## J  EFFECT OF LEARNING RATE AND BATCH SIZE

Figure 12: AUROCs obtained with an ensemble of WRN-28-10 models, as the initial learning rate and the training batch size are varied. For this we used the hardest setting, CIFAR100:0-50 as ID, and CIFAR100:50-100 as OOD.

## K  MEDICAL OOD DETECTION BENCHMARK

The medical OOD detection benchmark is organized as follows. There are four training (ID) data sets, from three different domains: two data sets with chest X-rays, one with fundus imaging and one with histology images. For each ID data set, the authors consider three different OOD scenarios:

1. Use case 1: The OOD data set contains images from a completely different domain, similar to our category of easy OOD detection settings.

2. Use case 2: The OOD data set contains images with various corruptions, similar to our category of hard covariate shift settings.

3. Use case 3: The OOD data set contains images that are not seen during training due to various selection biases, similar to our category of novel class settings.

The authors evaluate a number of methods on all these scenarios. The methods can be roughly categorized as follows:

1. Data-only methods: Fully non-parametric approaches like kNN.

2. Classifier-only methods: Methods that use a classifier trained on the training set, e.g. ODIN (Liang et al., 2018), Mahalanobis (Lee et al., 2018). RETO falls into this category as well.

3. Methods with Auxiliary Models: Methods that use an utoencoder or a generative model, like a Variational Autoencoder or a Generative Adversarial Network. Some of these approaches can be expensive to train and difficult to optimize and tune.

We stress the fact that for most of these methods the authors use (known) OOD data during training. Oftentimes the OOD samples observed during training come from a distribution that is very similar to the OOD distribution used for evaluation.

For exact details regarding the data sets and the methods used for the benchmark, we refer the reader to the paper Cao et al. (2020).

We did not evaluate RETO on the histology image data set due to resource limitations; the data set is much larger than the others.

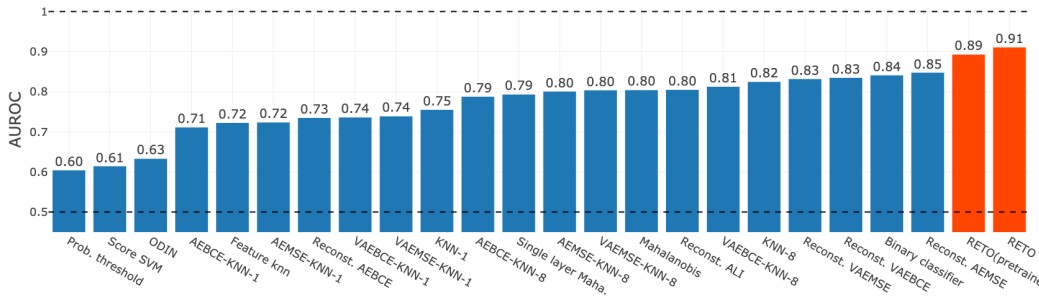

Figure 13: AUROC averaged over all scenarios in the medical OOD detection benchmark (Cao et al., 2020). The values for all the baselines are computed using code made available by authors of the paper Cao et al. (2020). Notably, most of the baselines assume oracle knowledge of OOD data at training time.

In Figures 14, 15, 16 we present AUROC and AUPR (Area under the Precision Recall curve) for RETO for each of the training data sets, and each of the use cases. Figure 13 presents averages over all settings that we considered, for all the baseline methods in the benchmark.

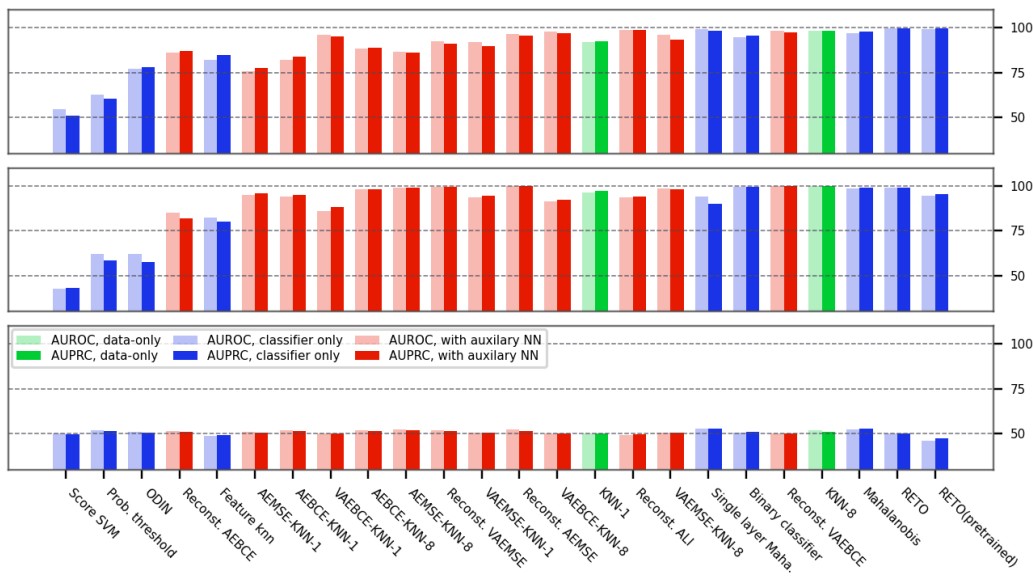

Figure 14: Comparison between RETO and the various baselines on the NIH chest X-ray data set, for use case 1 (top), use case 2 (middle) and use case 3 (bottom).

## L   TWO-SAMPLE TEST FOR OOD DETECTION

Distinguishing between ID and OOD samples can be cast as a two-sample hypothesis test, with $H_0 : x \in \text{supp}_P$ and $H_1 : x \notin \text{supp}_P$, for a sample $x$. The various OOD detection methods differ in their choice of the test statistic. For approaches that use ensembles of classifiers, the test statistic should reflect the belief that the models have similar outputs on ID samples, and disagree on OOD samples. For example, Lakshminarayanan et al. (2017) propose averaging the softmax outputs of all

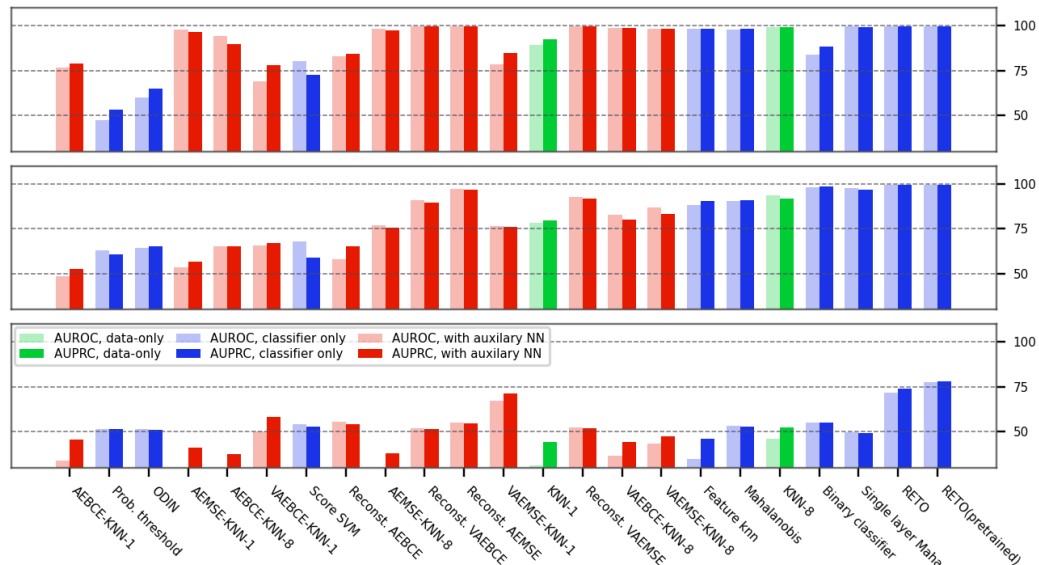

Figure 15: Comparison between RETO and the various baselines on the PC chest X-ray data set, for use case 1 (top), use case 2 (middle) and use case 3 (bottom).

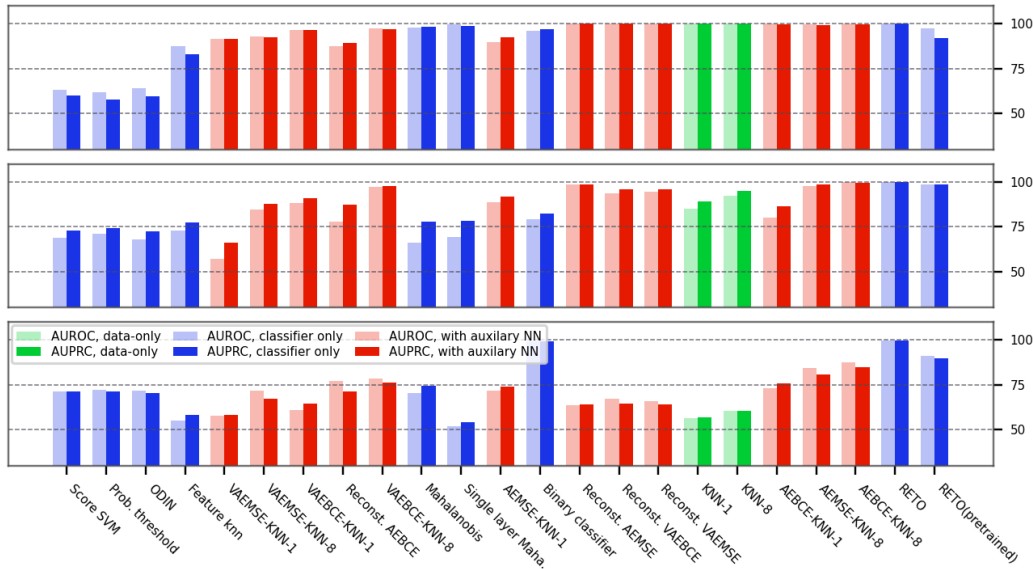

Figure 16: Comparison between RETO and the various baselines on the DRD fundus imaging data set, for use case 1 (top), use case 2 (middle) and use case 3 (bottom).

the models in the ensemble and then taking the maximum or the entropy of the resulting probability vector as the test statistic. For a $K$-model ensemble and an input $x$ this can be written as follows:

$$T_{\text{max-p}}(x) := \max_{i \in [C]} \frac{1}{K} \sum_{k=1}^{K} (f_k(x))_i, \text{ with } f_k(x) \in \mathbb{R}^C \text{ the softmax output of the k}^{\text{th}} \text{ model}$$

Averaging the softmax vectors loses some information about the model predictions, because different initial probability vectors can map to the same averaged vector. In our approach, models are more uncertain on ID samples than on OOD samples, which can make the averaged softmax vector

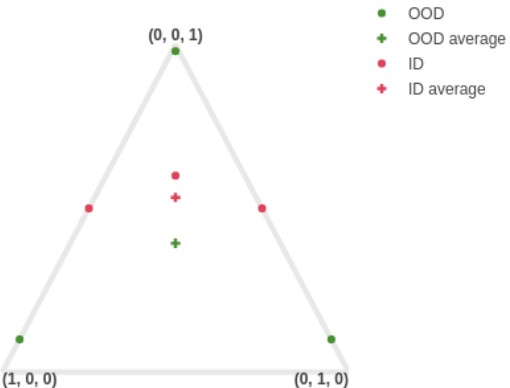

Figure 17: Averaging the probability outputs of the models in an ensemble can lead to catastrophic information loss, in some cases. The softmax vectors of a 3-model ensemble are represented on the 2D probability simplex. For an OOD sample, each model predicts with high confidence the arbitrary label it has seen during training. For the ID sample, the models predict the correct class with moderate confidence. Therefore, the average probability vectors for the ID and the OOD sample are close, which can make it hard to distinguish between them.

fall at the same location for an ID and an OOD sample. This makes it impossible to distinguish between the two, as illustrated in Figure 17.

For neural network ensembles, following a standard training procedure of minimizing the cross-entropy loss leads to models that make confident predictions on both ID and OOD samples, as shown by Hein et al. (2019); Maennel & Ţifrea (2020). Consequently, the information lost through averaging is not causing any issues: on ID samples, the models will tend to give the same prediction, while on OOD samples the models tend to disagree, giving different predictions with high confidence.

However, in our case, because of early stopping, the training process is halted at different stages for test ID and test OOD samples, as indicated in Figure 5.

Recent papers like Shwartz-Ziv & Tishby (2017); Chen et al. (2019) analyze the dynamics of optimizing the cross-entropy loss with SGD. They suggest that there might exist two stages: one in which a good decision boundary is found, and another in which the margin is increased between the representations of inputs from different classes. It is this second stage that also leads to overconfident predictions on both ID and OOD samples. Thus, early stopping may cause the models to be more uncertain on test ID samples than on test OOD. This is indeed confirmed in Figure 18.

To avoid the problem of information loss described previously, we compute the pairwise total variation distances between the softmax outputs of the models in the ensemble, and we take the average of these distances as our test statistic:

$$T_{\text{avg-TV}}(x) := \frac{2}{K(K-1)} \sum_{\substack{i,j=1 \\ i<j}}^{K} d_{\text{TV}}\left(f_i(x), f_j(x)\right)$$

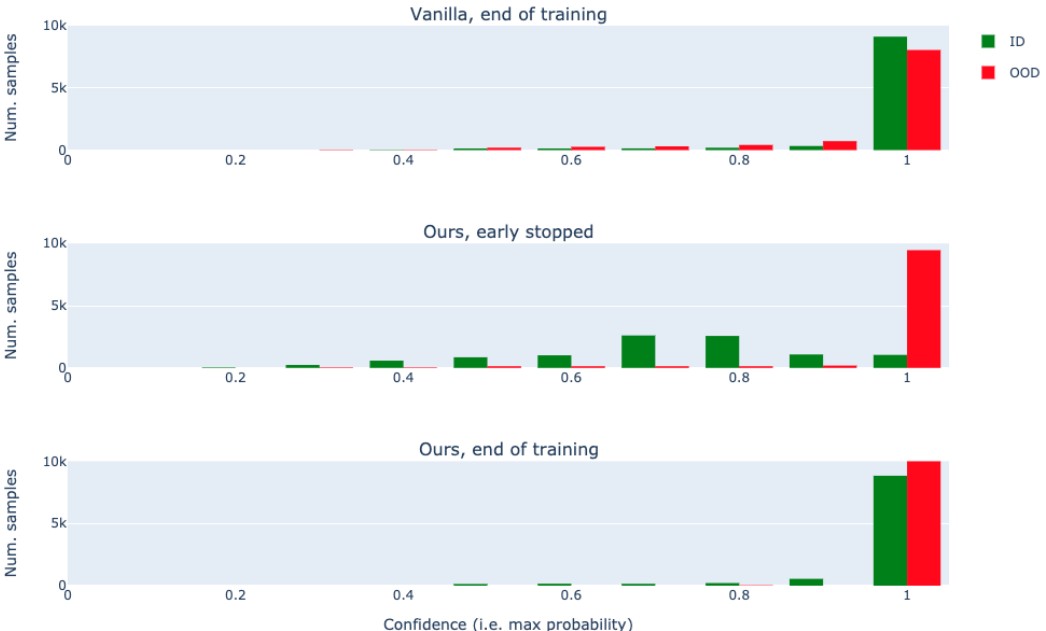

Figure 18: Distribution of confidence for a model trained on the training set alone (**Top**) and a model trained on both the training set and the arbitrarily labeled test set, with early stopping (**Middle**) and after 100 epochs (**Bottom**). The models in the vanilla ensemble are confident on both ID and OOD samples. The model trained on training+test is only confident on OOD data early during training.

