# OpenReview forum: "Learn what you can't learn: Regularized Ensembles for Transductive out-of-distribution detection"
_ICLR.cc/2021/Conference — Reject_

### Official Review · AnonReviewer4 · 2020-10-28
**Good performance for out of distribution (OOD) detection.**

**Rating:** 8
**Confidence:** 3

**Review:**

The problem of good predictive uncertainty-based out of distribution (OOD) detection is essential for classification systems to be deployed in safety-critical environments. The authors present a method RETO that achieves state-of-the-art performance in a transductive OOD detection setting. Like other predictive uncertainty-based approaches RETO can ultimately be used downstream on problems like active learning or abstaining on OOD samples in combination with selective classification.

Benchmark data such as CIFAR, SVHN, and MNIST are used to compare conventional and proposed baseline methods, such as k-NN, Vanilla Ensembles OE, Mahal, Mahal-T, and MCD. Experimental results, including those of supplemental materials, show that the proposed method provides good accuracy while reflecting the hardness of the task.

However, there is not enough discussion on how the proposed method can achieve such a high level of accuracy compared to the conventional methods; early stopping is used in RETO, but is it promised to reproduce the same level of performance in other tasks?

---

> ### Author Response · Authors · 2020-11-22
> **Response Reviewer 4**
>
> We thank Reviewer #4 for the positive assessment of our work.
>
> **Could early stopping be useful for other tasks?** For our approach, the signal in the correctly-labeled training data makes it difficult to learn the arbitrary label on the test ID samples. Conversely, the arbitrary label is usually learned much faster on OOD points, as shown in Figure 4 in our manuscript. By stopping early, we can capture an iteration when only test OOD samples are fit. We believe that this phenomenon can be exploited further for transductive classification, in contexts where samples from one class are abundant, and leave a thorough investigation of that direction for future work.
>
> **Why RETO outperforms the baselines.** We have tried to formulate clear arguments in Section 3 of the revised manuscript. Please also see our general reply.

---

### Official Review · AnonReviewer3 · 2020-10-28
**Simple method achieving impressive empirical results with some potential for improvement w.r.t. clarity of experimental procedures**

**Rating:** 6
**Confidence:** 2

**Review:**

I think the authors did a great job with the overall structure of the manuscript. It is great to see so many figures that illustrate the different settings, also it really helps to have the main research questions exposed right in the corresponding paragraph.

One superficial thing that could be improved is that often times I found the mathematical notation to be not very ‘convenient’ in that it wasn’t really necessary for a derivation or the like, but it made it really difficult to read the text, as it required jumping back and forth between the occurrences of a term and its definition, while the actual meaning of a term wouldn’t have taken much more space than its mathematical counterpart.

A minor thing: the authors use the abbreviation RETO in the abstract without mentioning the full name.

Figure 1 is great, I really appreciate these types of figures and I believe these comparisons are a great contribution to the community. Yet without details on the experimental settings that led to these results it is rather difficult to assess the value of this contribution. In particular as the figure is a central part of the motivation of the proposed approach, it would be great to at least reference the experimental details in Appendix A.1 - but even after having found this information (which shouldn’t be hidden from the reader for such essential parts), it remains a bit unclear how exactly the parameters were chosen and which experiments were performed exactly.

One of the main strengths of the paper is the extensive experimental validation. The authors did a great job at comparing with a variety of different methods.

The authors state that “For simplicity, we consider a noiseless setting, i.e. we assume that there exists a deterministic function f*”. It is perfectly fair to make such an assumption, but it is also important to keep in mind that in many real world application scenarios, especially when ground truth training data was acquired in an automated fashion or in crowdsourced annotation, label noise is not unlikely to violate that assumption. So it could help, for the assessment of the manuscript, to mention when this assumption would be violated in real world settings and how the proposed approach would be affected by such an assumption or a violation thereof.

The authors also state that “Empirically, this assumption [generalizability] seems to hold true for trained deep neural networks” and refer to Theisen 2020 - I might be missing something, but that reference is focussing on defining the “typical” rather than “worst” case, in particular for linear methods. In light of other findings like those that highlight that Neural networks will learn even noise labels perfectly fine, see e.g. Zhang et al https://arxiv.org/abs/1611.03530 it would be great to substantiate that claim a bit more.

The authors state that “Yu & Aizawa (2019) consider a similar transductive setting” and argue that that approach has the disadvantage that "the method produces indistinguishable outputs on ID and OOD samples”. I’m not sure I understand why that distinction is an advantage, after all the ultimate goal would be to perform well on target test data, independent of whether that data is ID or OOD.

A major limitation of the experimental validation is that bayesian methods are excluded from the evaluation. I have to admit I am not the most bayesian person I know, and I really enjoyed reading the Ovadia et al. 2019 NIPS paper, referenced by the authors, and I agree that this is a fair point, but after all this sort of transductive setting could be considered as one of the main strengths of bayesian/probabilistic networks. I would have expected to see at least one method from that field in the comparisons.

Overall the experimental results are very impressive. Yet there are some methodological details that could be clarified, next to the points listed above. Most importantly it felt that the hyperparameters of all methods were chosen in a bit of an ad hoc fashion. I think it would substantially improve the results to employ standard practices for the hyperparameter optimization for all methods, even just grid search or random search would be fine.

---

> ### Author Response · Authors · 2020-11-22
> **Response Reviewer 3**
>
> Firstly, we would like to thank Reviewer #3 for appreciating our writing style and contributions and for providing constructive feedback to our work. We have tried to incorporate their suggestions in our manuscript.
>
> **Figure 1 hiding too many details:** Thank you for the comment, we agree that even though the plot is easier to parse, it does not give a nuanced enough sense of our contributions. As a result, we chose to add what we believe is a more informative, yet still eloquent, plot in which we show how the OOD detection performance changes for various methods as the tasks increase in difficulty, ranked by RETO performance.
>
> **Hiding/Missing details on the experimental hyperparameter choices for baselines vs. RETO:** Thank you for pointing out that this was not stated clearly enough in the main text, we have now moved a discussion from Appendix A to Section 4.2. and are now stating very clearly how we choose the hyperparameters. We want to stress in particular that our hyperparameter choices do not depend on the unseen OOD data. In particular, for RETO, we chose hyperparameters such that one model trained on the training set achieves state-of-the-art test accuracy in the *training (ID) distribution* with the respective architecture (e.g. ResNet18, WideResNet, VGG etc). For all the baselines we considered, we used the hyperparameters suggested by the authors for the respective training datasets (CIFAR10/CIFAR100, SVHN, Tiny Imagenet) and made sure we can indeed reproduce the reported results. We have also added experiments that show the stability of our approach with respect to the choice of hyperparameters in Appendix J. In the interest of transparency and reproducibility, we are also making our code publicly available at [1].
>
> **Missing Bayesian baseline as major limitation:** Indeed, we agree that comparing with Bayesian baselines would be good to complete the picture. Hence, after the submission deadline we have also conducted experiments for Deep Prior Networks [2] that we now added to Table 1 for a more comprehensive comparison with prior work.
>
> **‘Indistinguishable outputs on ID and OOD’:** The problem we are tackling in this work is that of out-of-distribution detection, namely we want to identify samples that are ‘unlike’ the ones in the (ID) training set. Having said that, we have tried to make the paragraph indicated by Reviewer #3 clearer in the new version of the manuscript.
>
> [1] - http://bit.do/reto-code-iclr
>
> [2] - Malinin et al, Predictive Uncertainty Estimation via Prior Networks, https://arxiv.org/abs/1802.10501

---

### Official Review · AnonReviewer1 · 2020-10-28
**A technique for discriminating between two known distributions, incorrectly claimed to be OOD detector.**

**Rating:** 6
**Confidence:** 3

**Review:**

The paper proposes a technique to detect OOD data using disagreement among an ensemble of classifiers. It assumes a transductive setup where OOD data is available during training time.


1. The application setting for OOD as stated in the paper is incorrect:

Abstract: "For settings where the test data is available at training time.."

Introduction: "..Most prior work approaches OOD detection inductively, trying to infer OOD samples after only observing the (ID) training data,.."

The main property defining OOD data is that it is never seen during training. The problem fundamentally is that of detecting unknown-unknowns. This is what makes the problem hard and sets it apart. The proposed approach violates this and hence is not justified as an OOD detection algorithm. The paper text (including title) should be changed.

The proposed technique should instead be positioned as one that helps in discriminating between two known distributions. In that respect, the baseline/competitor algorithms should be selected appropriately. The current choices (except Malalanobis-T) are biased towards the proposed algorithm.

2. Section 2, ID data.: It is assumed for simplicity that a deterministic function f* maps ID data to correct labels. However, this assumption looses generalizability. Considering that real world data has a lot of labeling noise, most such functions as f* would end up memorizing the train data and would have bad performance on ID data in the test set. The paper is not clear what would happen if the train data is noisy.

3. Section 2, Transductive OOD Detection: Here the assumption is that there is lot more correctly labeled train data than test data. The basis for this assumption is not clear and clearly limits the applicability. In most transductive cases, the opposite is true: there is more unlabeled data than labeled, and semi-supervised techniques are used to reduce human labeling effort.

4. Section 2.2, Regularized ensembles: "..points in T for which at least two models in the..." -- Does that not make the model fragile? In case the ensemble has 20 members and only one member disagrees with 19 others, would it be justified to label the data as outlier/OOD?

5. Section 2.2: "...our method indicates that using the regularized function class suffices for many hard OOD detection scenarios." -- This is an overstatement considering that results were presented on only a couple of datasets (CIFAR variants, SVHN)

6. Section 3.2, Statistical test for OOD detection: "The null hypothesis is accepted for high values of Tavg-TV." -- I am missing something here: if there is a lot of disagreement among ensembles, then d_TV would be high and as a result for such data Tavg-TV would be also high. Hence, high value of Tavg-TV should probably result in rejecting of null hypothesis instead of accepting.


============================
Update after going through the updated paper and discussions
---------------------------------------------
The paper adds better illustrations in the revised paper to explain the technique and the approach seems to be effective even though it is simple. Given the approach, 'transductive' in the title looks okay.

Figure 1, Figure 3, and Figure 4 in the current version suggest that the one-class SVM/SVDD techniques [1, 2] would be important to compare against. Popular anomaly/outlier detection algorithms Isolation Forest [3], LOF [4] would also be relevant here. The paper can be strengthened with these additional baselines. RETO has at least one short-coming over the outlier detector techniques: it needs to wait some time until it can collect enough test data.

[1] Lukas Ruff, Robert A Vandermeulen, Nico Görnitz, Lucas Deecke, Shoaib A Siddiqui, Alexander Binder, Emmanuel Müller, and Marius Kloft. Deep one-class classification. In ICML, volume 80, pp. 4390–4399, 2018.

[2] Lukas Ruff, Robert A Vandermeulen, et. al., Deep Semi-Supervised Anomaly Detection, ICLR 2020.

[3] Liu, Fei Tony; Ting, Kai Ming; Zhou, Zhi-Hua (December 2008). "Isolation-based Anomaly Detection". ACM Transactions on Knowledge Discovery from Data

[4] Breunig, M. M.; Kriegel, H.-P.; Ng, R. T.; Sander, J. (2000). LOF: Identifying Density-based Local Outliers. Proceedings of the 2000 ACM SIGMOD International Conference on Management of Data. SIGMOD.

---

> ### Author Response · Authors · 2020-11-22
> **Response Reviewer 1; 1/3**
>
> We thank the reviewer for all their comments on different parts of our manuscript. Thanks to these comments we have made significant efforts to clarify formulations that make the paper easier to follow for the new reader. In what follows we mainly want to argue the transductive OOD setting warrants its name and hope to be able to convince R1 of that or are happy to hear concrete counter arguments.
>
> **Completely different setting than other OOD works:**
> - (1) [ The main property defining OOD data is that it is never seen during training. ]
> - (2) [ The problem fundamentally is that of detecting unknown-unknowns. This is what makes the problem hard and sets it apart. ] The proposed approach violates this and hence is not justified as an OOD detection algorithm. The paper text (including title) should be changed.
>
> Regarding (1) OOD data defined as not seen during training: (this paragraph is copied from the general rebuttal comments for everyone)
> We would like to argue that the problem of classical OOD detection is as follows: for given test points, determine for which samples there is no information in the *labeled* training data to label them. These we call OOD. And indeed, we want to do so *without* having explicitly seen any examples for OOD points labeled as such. So we would agree with a modification of the reviewers sentence as follows (and hope that the reviewer also agrees): The main property defining OOD data is that no samples of that set have been seen with a label (labeled as OOD or their class label) during training. And yes, this is exactly our setting, the same as for the baselines. Nonetheless there are indeed differences in the training between transductive and inductive OOD settings which we lay out first before arguing when the transductive setting can be important in practice.
>
> Regarding hardness (2) compared to the usual (inductive) setting: We want to stress again that our method, RETO, does not see OOD data (labeled as such) during training. The only labels that are available are class labels in the ID training set and we do not know which of the test samples are OOD. However, transductive OOD detection is indeed slightly “easier” but only in the sense that it can use additional resources. That is, when we’re given the test data, we still have access to the training data and computational resources to tune our OOD detection procedure (this is the dual perspective to having access to the test set during training time, as the reviewer put it). The inductive setting, by definition, cannot do any tuning during test time (although in reality, some of the inductive baselines effectively use the exact same OOD data distribution during training for hyperparameter tuning [2], [3]. This is often not explicitly stated in the description of the method.
>
> We want to emphasize, however, that we were rather open about transductive OOD detection being a different, slightly “easier” setting and we make this very explicit in the italic question on page 2. The goal of this paper is indeed to explore *how and how well* we can leverage the additional resources in this supposedly “easier” setting, especially in OOD scenarios where SOTA inductive methods do not perform well. The answer to the question is not at all obvious and the main contribution of the paper.
>
> In the revised manuscript we also added more vivid practical scenarios for when transductive OOD settings are indeed valuable. We have also tried to argue in a bit more detail why the transductive OOD setting still warrants the name OOD and why it’s worth studying. Both of these changes can be found in paragraph 4 of the introduction and Section 2.1.

---

> > ### Author Response · Authors · 2020-11-22
> > **Response Reviewer 1; 2/3**
> >
> > **Incorrect choice of baselines for comparison:** We do indeed compare our approach with baselines with the same setting: Maximum Classifier Discrepancy [1], and a transductive variant of the Mahalanobis baselines (Mahal-T in Table 1). We included inductive baselines as well, as the goal of the paper was indeed to compare how much we can improve using the extra information in the transductive setting. And, as mentioned above, some of these inductive baselines indeed use OOD data, which is actually making it harder for RETO to improve upon.
> >
> > **Transductive OOD setting requires large labeled training data is a limitation:** First we want to note that our transductive OOD method and setting does not require *more* correctly labeled training data than any other inductive OOD method. Again, the reason is that OOD detection usually comes in when generalization is good (there are few known unknowns), but we are still prone to see unknown unknowns. Hence, it is not a restriction, but a fact that enough labeled training data is available for in-distribution generalization. Otherwise we would need to find a different model or more in-distribution data (labeled or unlabeled) first, to predict better in-distribution.
> >
> > **Transductive OOD detection improves upon inductive OOD detection for large labeled training set, whereas transductive classification improves upon inductive especially for small labeled training set:** Thank you for the interesting observation. The regime where transductive *OOD detection* improves upon inductive OOD detection is indeed different from the transductive *classification* case. A main reason is that transductive OOD detection is fundamentally different from transductive classification and hence intuitions do not transfer (also a reason why it is not obvious how well one can leverage the easier transductive setting discussed above). We now try to give intuition why the regimes of advantage are rather different.
> >
> > In transductive classification, the labels in the (small) training set are informative for the desired labels in the unlabeled test set (large). The unlabeled test set can then be used to train a good embedding / representation and the (few) labels from the training set can then propagate through. The success of transductive methods in that setting hence heavily relies on the assumption that both training and test set are related, for instance the labeled and the unlabeled samples come from the same distribution.
> > Since there is no prior knowledge (in terms of labels) of any OOD samples in the training set, as noted in our revisited definition of OOD above, we can only leverage the information in the (ID) training set to separate the ID and the OOD samples in a test batch. The lack of information about the OOD distribution is counterbalanced by more ID training data.
> >
> > **Fragility of ensembles if only 2 models out of 20 disagree “..points in T for which at least two models … disagree”:** We agree that the phrasing was misleading and have made this statement more precise in the revised manuscript. The disagreement statistic we propose is in fact smooth and captures how often models produce different predictions on test OOD data compared to test ID data. The decision to flag a sample as OOD is then determined by comparing the test statistic to a threshold (which we can choose depending on the costs of false positives or false negatives).
> >
> > **Overstatement to say that RETO improves on “many” hard OOD detection scenarios:** Most recent OOD detection works show near-perfect performance on a number of commonly used data sets (e.g. CIFAR10 vs SVHN). Instead we focus on more difficult settings, e.g. first 5 classes of SVHN as ID vs last 5 classes as OOD, ImageNet vs ObjectNet [4], CIFAR10 vs Corrupted CIFAR10 [5] etc. We find that previous OOD detection approaches do not perform well on these settings. We show RETO’s better performance compared to the baselines for different OOD scenarios in Figure 1. In addition to these settings, we have added experimental results obtained on more realistic scenarios, namely on a recently proposed medical OOD detection benchmark [6] in section 4.
> >
> > **Typo in statistical test for OOD detection:** Thank you for catching the typo, the null hypothesis is indeed rejected for high values of Tavg-TV. We have corrected the statement in the manuscript.

---

> > > ### Author Response · Authors · 2020-11-22
> > > **Response Reviewer 1; 3/3**
> > >
> > > [1] - Yu et al, Unsupervised Out-of-Distribution Detection by Maximum Classifier Discrepancy, https://arxiv.org/abs/1908.04951
> > >
> > > [2] - Liang et al, Enhancing The Reliability of Out-of-distribution Image Detection in Neural Networks, https://arxiv.org/abs/1706.02690
> > >
> > > [3] - Lee et al, A Simple Unified Framework for Detecting Out-of-Distribution Samples and Adversarial Attacks, https://arxiv.org/abs/1807.03888
> > >
> > > [4] - Barbu et al, ObjectNet: A large-scale bias-controlled dataset for pushing the limits of object recognition models, https://papers.nips.cc/paper/2019/hash/97af07a14cacba681feacf3012730892-Abstract.html
> > >
> > > [5] - Hendrycks et al, Benchmarking Neural Network Robustness to Common Corruptions and Perturbations, https://arxiv.org/abs/1903.12261
> > >
> > > [6] - Cao et al, A Benchmark of Medical Out of Distribution Detection, https://arxiv.org/abs/2007.04250

---

### Official Review · AnonReviewer2 · 2020-11-02
**strong experimental results, but more rigorous justification and further details are required**

**Rating:** 4
**Confidence:** 4

**Review:**

This paper proposes a new approach to detect out-of-distribution (OOD) examples in the transductive setting. The idea is to train an ensemble of models that fit the in-distribution (ID) data well, but disagree with each other on OOD examples.

Pros
+ Extensive experiments are conducted to compare the proposed alogrithm with a number of baselines.  The proposed algorithm seems to significantly out-perform the baselines.
+ The paper is generally well-written and easy to follow.

Cons
- The concept of the F-detectable OOD set OOD(P, F) is somewhat misleading: the name suggests that it is a set of OOD examples, while in fact this set is not necessarily a subset of OOD examples, because when there are multiple empirical minimizer for P, they may disagree on some ID examples.
- A key assumption (the empirical minimizers of F generalises well) is justified by a miscitation: the paper states that Theisen et al. 2020 shows that the assumption seems to hold true for trained deep neural networks, while that paper only discusses overparametrized linear classifiers.
- Another key assumption (Assumption 2.1) seems rather strong and unlikely to hold in practice, and its validity seems to be overstated. Specifically, the assumption assumes an empirical minimizer of the training set augmented with ID test examples labelled using an arbitrary label is an empirical minimizer on the training set. This will imply that the ID test examples are not important in the underlying data distribution. While the papers states that "In Section 3 we provide empirical evidence which shows that Assumption 2.1 holds in many practical scenarios", there is no direct support for the assumption in Sec 3.
- The paper states that OOD(P, \tilde{F}) \subset OOD(P, F), because \tilde{F} \subset F. This is not necessarily true, because the empirical minimizers in \tilde{F} can disagree on more ID examples than the empirical minimizers in F.
- Fig. 4 is used to justify that early stopping is an effective regularization method. However, the figure shows that at around epoch 35, there is a sharp drop of the validation accuracy, and early stopping may stop training here, making the network underfit.
- The paper doesn't seem to describe how the test statistics are converted into predictions of whether the examples are OOD. This is important for reproducibility and is needed for justifying the soundness of the algorithm.

Minor comments
- Write down ID in full when it's used the first time
- Fig. 2 is not mentioned in the main text. In addition, the caption doesn't seem to match the figure.
- Fig. 3: what does zoro bias mean here?
- Alg. 1: y_{1}, ..., y_{K} have already been used a the training set labels.
- The "two sample test" doesn't really use two samples.

**Post-rebuttal**

The revised version removed questionable or invalid notations/assumptions/justifications, and there are some defences for previous formal justifications in the rebuttal. IMHO, the rebuttal raises further concerns about the technical quality, and the paper still requires stronger justification for acceptance. I'll lower my score instead.

Specifically, I find the rebuttal generally confusing, and I disagree with various points in the rebuttal/revised version.

- Condition 3.1 doesn't seem right: by definition, a classifier in
 has a misclassification probability of at most  on the ID data, but Condition 3.1 states that the misclassification rate is ? Importantly, I don't see why this condition is needed as there is no justification on how it is connected to the disagreement test. In addition, "As a consequence, with very high probability 1 − (1 − δ)s we cannot fit a set of s random i.i.d. in-distribution points with the wrong label" is quite vauge and doesn't seem right either.
- "According to the definition, there exists a class of functions F that is complex enough such that OOD(P, F) is the complement of the support of the training distribution.": I don't think the definition implies this. In addition, if OOD(P, F) is the complement of the training distribution, then ID examples not in the training distribution are included in OOD(P, F).
- "If we defined OOD(P_n, F) with P_n the empirical distribution, then the reviewer would be correct and indeed this set could contain ID samples.": in theory, OOD(P, F) CAN contain ID samples, unless additional assumptions are made.
- "two-sample test": while I think this is a minor issue to call the test a two-sample test, I still don't think it agrees with the standard usage of the term (which means two random samples are used).
- "one may simply pick the time point with the highest validation accuracy, after training for a fixed number of epochs": I doubt that this can be called "early stopping". So I don't think the response addresses my question regarding Fig. 5 (previous Fig. 4). In addition, since "early stopping" is used for regularization, this makes it questionable whether "regularization" is indeed needed.
- The test statistic is included in the main paper now, but how about the threshold? In rigorous statistical testing, the threshold can be rigorously calculated, while in this case, it is not clear how the threshold is set.

---

> ### Author Response · Authors · 2020-11-22
> **Response Reviewer 2**
>
> We thank the reviewer for their constructive suggestions that have convinced us to omit some formalisms that were not necessary to convey our main story and contributions. We hope our changes to the manuscript, in particular regarding Sections 2 & 3, address most of the concerns that were raised and make our points clearer and better justified.
>
> **Definition of OOD(P,F), former Ass. 2.1. (now Condition 3.1.):** Our main empirical contribution was to improve OOD detection for very expressive models, like deep neural networks. In the context of highly-overparametrized models, the distinction between F-detectable and undetectable OOD samples (which we originally introduced for more generality) is not very insightful: the undetectable OOD set would be empty for our use cases, since functions parametrized by neural networks are complex enough to represent perfectly the support of the training distribution. Hence we have removed the definition altogether and all the necessary symbols we introduced for this purpose. Please see our discussion in the general comments regarding specific changes. We hope the current version clarifies most of the issues. However here are answers to the comments regarding the original manuscript for completeness:
>
> **OOD(P,F) including ID points:** We first want to clarify that OOD(P,F) per our former definition was defined as the F-detectable OOD set with respect to minimizers of the *population* error (i.e. infinite sample regime), indicated by P. According to the definition, there exists a class of functions F that is complex enough such that OOD(P, F) is the complement of the support of the training distribution. If we defined OOD(P_n, F) with P_n the empirical distribution, then the reviewer would be correct and indeed this set could contain ID samples.
>
> **OOD(P, \tilde{F}) \subset OOD(P, F) being incorrect:** This was indeed another inconsistency and a remainder from the discussion which relied on the noiseless assumption. In that case we assumed f* is in \tilde{F} which makes this inclusion valid (again, in the population setting).
>
> **Drop in validation accuracy in Figure 4:** Our method builds upon classifiers that already generalize well on the training distribution. That means we would wait until training and validation accuracy are as high as the classifiers can achieve without the arbitrary labels before activating the early stopping rule. In practice, one may simply pick the time point with the highest validation accuracy, after training for a fixed number of epochs. For transparency, we have now adjusted  the description of our early stopping procedure to be more explicit on this issue in the main text of the paper (in section 4.3).
>
> **Alg. 1: y_{1}, ..., y_{K} have already been used as the training set labels:** Our method does indeed assign one of the training set labels to the whole test set. We hope our revised manuscript makes this more clear.
>
> **Two-sample test not using two samples:** The OOD detection problem in the recent ML literature is reminiscent of a two-sample statistical hypothesis test [1]: under the null hypothesis, a test sample lies in the same support as the training data. To assess the quality of the statistical test, we employ evaluation metrics that are established in the OOD detection literature, like the area under the ROC curve, or the false positive rate at a fixed true positive rate (which we set to be 95%). Moreover, in the interest of reproducibility, we make our code available [2].
>
> **How to use the test statistic for OOD detection:** Some of the most powerful statistical tests (see e.g. Neyman-Pearson Lemma, likelihood ratio tests [3]) rely on comparing a test statistic with a threshold which depends on the desired significance level / allowed false positive error. The samples are then flagged as H_0 or H_1 accordingly. We do the same with our statistic. To have a complete picture of the procedure in one place, we have now streamlined the presentation and moved the description of the test statistic and how it is used for OOD detection, right after the description of the algorithm in Section 2.
>
> **Figure 3:** We removed it since we removed the definition of OOD(P,F). Zero bias referred to the constant bias term of linear classifiers w^T x + bias. Figure 3 a) illustrated the Undetectable OOD for this hypothesis class.
>
> [1] - https://en.wikipedia.org/wiki/Two-sample_hypothesis_testing
>
> [2] - http://bit.do/reto-code-iclr
>
> [3] - https://en.wikipedia.org/wiki/Neyman%E2%80%93Pearson_lemma

---

### Author Response · Authors · 2020-11-22
**General comments 1/2**

We thank all the reviewers for appreciating our work and for providing constructive comments. Based on these we were able to restructure and rephrase large parts of the paper for the sake of clarity. We want to note that we did not add much new content, but on the contrary rather removed some formalisms that seemed more distracting than helpful. In the following, we first want to comment on the reviewers’ major concerns and some resulting changes and clarifications in the manuscript.  We then move on to address individual comments.

**[Especially R1] Transductive OOD setting: its validity and importance.**

We would like to argue that the problem of (classical) OOD detection is as follows: for given test points, determine for which samples there is no information in the *labeled* training data to label them. We call these “test OOD points”. And indeed, we want to do so *without* having explicitly seen any examples for OOD points labeled as such. So we would agree with a modification of the reviewers sentence as follows (and hope that the reviewer also agrees): The main property defining OOD data is that *no samples of that set have been seen with a label (labeled as OOD or their class label)* during training. And yes, this is exactly our setting, the same as for the baselines. Nonetheless there are indeed differences in the training between transductive and inductive OOD settings which we lay out first before arguing when the transductive setting can be important in practice.

In brief, the difference between most other baselines (inductive) and our (transductive) setting is that the inductive setting doesn’t allow for tuning the method during test time. The inductive OOD detector has to be trained beforehand, whereas the transductive OOD detection method can be fine-tuned using the additional information captured in a batch of unlabeled test samples. We want to stress that, by design, neither setting has access to OOD data labeled as such. The test batch contains *both* ID and OOD data (not *only* OOD data). Since no labeled OOD data is known, the outliers in the test batch are unknown-unknowns.

Moreover, we argue that transductive OOD detection can be relevant in many practical settings: For example in situations where (online) inductive methods are not reliable enough but OOD detection is critical for damage/cost control, using a transductive OOD method (usable for a batch) can be worth the extra training effort even if that requires a small delay (see Covid example in new Section 2.1.). This is true for example when the cost for predicting OOD data could significantly surpass the cost of fine-tuning an ensemble of classifiers. We provide concrete examples of such situations in the introduction section of our paper and in Section 2.1.

**[R1, R2, R3] Main contributions vs. mathematically heavy former Section 2.** Our main contributions (agreed upon by most reviewers) consist of providing experimental evidence of hard OOD scenarios for neural networks and the proposition of a new method that is shown to improve the performance in all cases in the transductive setting. In contrast, with the mathematical formalism we merely wanted to include a rigorous definition of the (solvable, or F-detectable) OOD detection problem for *general* hypothesis classes as a path to derive RETO more formally.

Since our experiments and empirical contributions are exclusively for neural networks, we decided to only focus on overparameterized hypothesis classes in the entire introduction of the problem setting as well. However, for overparameterized classes, the entire complement of the training distribution support is detectable and hence this distinction between detectable/undetectable OOD sets is irrelevant. This allows us to remove most of the mathematical notation that caused some confusion. Furthermore we are also able to streamline the presentation of the paper much better: we now first present the transductive setting and the algorithm in Section 2, and leave all the motivation for why our method works to Section 3. Therein, we also minimized our usage of mathematical formalisms as much as possible for improved clarity.

---

> ### Author Response · Authors · 2020-11-22
> **General comments 2/2**
>
> We would now like to address specific comments that were mentioned by multiple reviewers and point to how we have incorporated them in the new version of the manuscript:
> - **[R3] Clarity of mathematical notation.** In the interest of clarity, we first describe the transductive OOD detection setting and the complete RETO procedure approach with some high-level motivation. In section 3 we then provide some more in-depth justification for why we think RETO performs well and the importance of regularization.
> - **[R2, R3] Assuming models with good generalization, Theisen citation.** We first want to clarify that this assumption was not really an additional one, but more like a restatement of our use case. In our use case, OOD detection becomes important when a model is about to be deployed - for which good generalization is a necessary condition. If a model has poor generalization, we cannot rely on its predictions on in-distribution (ID) data, so it is pointless to try and identify ‘unknown-unknowns’ for it and the data scientist should be more worried about finding a better model. Given that we have these models at hand, we can readily use them for our RETO ensemble, where good generalization helps control the false positive error (mistakenly flagging ID data as OOD). In practice we can ensure that the trained models are in this class by checking the validation accuracy. We have stated this more clearly in the paper, and have also removed the citation of Theisen et al. 2020, which, as correctly pointed out by Reviewers #2 and #3, only discusses the typically good generalization of overparametrized models in the linear case, though their study was motivated by the general case.
> - **[R2] Validity of Assumption 2.1. in practice (now Condition 3.1.).** Assumption 2.1. (Condition 3.1 in the revised version) is best interpreted as a desired property of the hypothesis class used to construct the ensemble and we have changed the manuscript to reflect this perspective more clearly. It is not a condition on the finite data sample, but rather on the complexity of the function class of the classifiers: the models are smooth enough that they *cannot* fit the ID test examples with an arbitrary label without sacrificing training and validation accuracy. We have now removed the formalism and stated the condition more cleanly which hopefully clarifies the doubts of R1 on this issue. Restricted function classes, for instance linear classifiers, satisfy this condition, as illustrated in Figure 2. Similarly, deep neural networks regularized with early stopping are also good at combating label noise (as shown both theoretically [1] and empirically [2]), thus satisfying our assumption.
> - **[R1, R3] Noiseless setting assumption.** We thank the reviewers for catching this - indeed, the assumption of no label noise in the training data is not necessary for RETO to perform well. It was in fact a remnant from one of our earlier attempts to formulate the OOD detection problem mathematically and we have hence removed it.
> - **[R4] Intuition for the better performance of RETO.** Thanks for the very good question. Compared to previous inductive methods, by tuning the OOD detection method after seeing a test batch, the transductive approaches use more information than inductive methods. However, it is not evident how to take advantage of this information to improve the detection of outliers. Compared to previous transductive work, the key for the stark improvements is that by adding regularization we are exploiting the gap between the complexity of functions that fit an arbitrary label on ID test samples vs the complexity of functions that fit arbitrary labels on OOD. We tried to crystallize this reasoning more in the new version of the manuscript. Building on prior theoretical and experimental works, we argue that for neural networks, early stopping can serve as an effective regularizer as the complexity is often reflected in the fitting “speed”.
>
> **[R2, R3] Algorithmic details and experiments.** In the new manuscript, we emphasize more clearly the RETO procedure (including the statistical test for anomaly detection) in section 2.2. In Appendix A we provide more details about the experiments, both for RETO, and the baselines, and show the stability of our method with respect to hyperparameter choices in Appendix J. We have extended our empirical analysis by adding a Bayesian method (as suggested by R3) and by evaluating RETO on a recently proposed medical OOD detection benchmark (more details in Appendix K).

---

### Decision · Program_Chairs · 2021-01-07
**Final Decision**

**Decision:**

Reject

**Comment:**

Although the rebuttal helped clarify the reviewers' confusion on notational confusion and the motivation of problem setup, all reviewers are still in a position of unable to championing the paper:
- the technical concerns by Reviewer 4 need to addressed
- the paper would have been stronger if baselines such as one-class classification / outlier detection have been compared
- the algorithm also has at least one short-coming over other techniques that it needs to wait some time until it can collect enough test data

We hope the reviews can help the authors strengthen the paper in the next revision.